# Biomimetic versus arbitrary motor control strategies for bionic hand skill learning

Hunter R. Schone [1,2,3,4] ✉, Malcolm Udeozor[1], Mae Moninghoff[1], Beth Rispoli [1], James Vandersea [5], Blair Lock[6], Levi Hargrove[7,8], Tamar R. Makin [2,9,10] ✉ & Chris I. Baker [1,10]

A long-standing engineering ambition has been to design anthropomorphic bionic limbs: devices that look like and are controlled in the same way as the biological body (biomimetic). The untested assumption is that biomimetic motor control enhances device embodiment, learning, generalization and automaticity. To test this, we compared biomimetic and non-biomimetic control strategies for non-disabled participants when learning to control a wearable myoelectric bionic hand operated by an eight-channel electromyography pattern-recognition system. We compared motor learning across days and behavioural tasks for two training groups: biomimetic (mimicking the desired bionic hand gesture with biological hand) and arbitrary control (mapping an unrelated biological hand gesture with the desired bionic gesture). For both trained groups, training improved bionic limb control, reduced cognitive reliance and increased embodiment over the bionic hand. Biomimetic users had more intuitive and faster control early in training. Arbitrary users matched biomimetic performance later in training. Furthermore, arbitrary users showed increased generalization to a new control strategy. Collectively, our findings suggest that biomimetic and arbitrary control strategies provide different benefits. The optimal strategy is probably not strictly biomimetic, but rather a flexible strategy within the biomimetic-to-arbitrary spectrum, depending on the user, available training opportunities and user requirements.

In an iconic scene in science-fiction cinema, Luke Skywalker (in *The Empire Strikes Back*) is shown examining his new bionic prosthetic hand. The device appears to, nearly perfectly, mimic a biological hand in its visual appearance and function. The control of the hand appears intuitive, such that Luke can immediately manipulate the individual digits with high dexterity. Today's technology is far from Skywalker's—even the most advanced bionic limbs are slow and operated using just a few degrees of freedom. Still, the design of artificial prosthetic hands has steadily innovated appearance and functionality to be increasingly similar to a biological hand: from sixteenth-century iron-clad hands with no manipulability, twentieth-century body-powered hook devices capable of simple grasping[1] to modern multi-gestural bionic hands that can be operated via electromyography (EMG) pattern-recognition control systems (Coapt Complete Control system; Ottobock Myo Plus;

[1]Laboratory of Brain and Cognition, National Institute of Mental Health, National Institutes of Health, Bethesda, MD, USA. [2]Institute of Cognitive Neuroscience, University College London, London, UK. [3]Rehab Neural Engineering Labs, University of Pittsburgh, Pittsburgh, PA, USA. [4]Department of Physical Medicine and Rehabilitation, University of Pittsburgh, Pittsburgh, PA, USA. [5]Medical Center Orthotics and Prosthetics, Silver Spring, MD, USA. [6]Coapt, Chicago, IL, USA. [7]Department of Physical Medicine and Rehabilitation, Northwestern University, Chicago, IL, USA. [8]The Regenstein Foundation Center for Bionic Medicine, Shirley Ryan AbilityLab, Chicago, IL, USA. [9]MRC Cognition and Brain Sciences Unit, University of Cambridge, Cambridge, UK. [10]These authors contributed equally: Tamar R. Makin, Chris I. Baker. ✉e-mail: schonehunter@gmail.com; tamar.makin@mrc-cbu.cam.ac.uk

for a review of available bionic prosthetic hands see ref. [2]). Driving much of the previous research and development in prosthetics, and the future trajectory of this industry, is the long-standing engineering ambition to design anthropomorphic artificial limbs: devices that look like and are controlled in the same way as the biological body, that is, devices that are biomimetic[3]. Biomimetic design has also driven the development of more invasive human–machine interfaces, such as artificial sensory feedback systems[4–10] and brain–computer interfaces[11]. Across these various approaches, biomimetic-inspired design in human–machine interfaces is predicated on the (largely untested) assumption that biomimetic devices potentially allow users to recruit pre-existing neural resources, supporting the biological body to assist device control, thereby enhancing device learning, generalization, sense of embodiment and automaticity. But are these assumptions that underlie biomimetic design valid?

If these assumptions are valid, we would expect the brain to integrate neural representations of external devices with the biological body to support an efficient recruitment of neural body resources. However, one growing body of evidence has suggested that this may not be feasible. Recent neuroimaging studies have found that individuals with extensive experience using a device as a hand replacement (prosthetic hands or expert grasping tools) neurally represent their devices less like a biological hand (that is, more distinct representations) as compared with novices[12,13]. So then, why should devices be designed to mimic the body if the brain does not seem to process even the most highly used, external devices in the same way as a biological body part? Moreover, considering the stark differences between biological and modern bionic limbs (for example, response time, dexterity, functionality, aesthetics, comfort or fit, weight, durability and sensory feedback), there are multiple ways in which biomimetic interfaces may actually introduce neurocognitive conflicts for users between pre-existing information or resources for the biological body and those for the artificial device[14]. Lastly, considering that most surveys of prosthesis users report high rates of prosthesis dissatisfaction and complete device abandonment[15,16], a critical re-evaluation of the research priorities driving development of these devices is warranted. In particular, it is vital to evaluate non-biomimetic control strategies that prioritize other design considerations (such as user requirements) over explicit biomimetics.

In this Article, we compared biomimetic and non-biomimetic motor control strategies directly while participants learned to operate a bionic hand, operated by an eight-channel EMG pattern-recognition system (Coapt; the most advanced commercially available system for controlling myoelectric bionic limbs). As a striking alternative to biomimetic control (as implemented for existing myoelectric technology), we incorporated an arbitrary (non-biomimetic) control strategy. On the basis of the neurocognitive assumptions underlying biomimetic design, this strategy should provide no direct benefits for the user. The primary rationale of the arbitrary strategy is to provide a contrast to biomimetic control. When choosing arbitrary gestures, we prioritized gestures that are not involved in typical object interaction but are otherwise easy to instruct, memorize, execute and replicate (for example, one finger to four fingers). The main idea is that this control strategy is moving away from natural movement and prioritizing considerations other than biomimetics. To test bionic hand skill learning, we trained non-disabled participants ($n = 40$) to use a wearable myoelectric bionic hand (Fig. 1). We assessed motor learning on multiple bionic hand skills across 4 training days (2–3 h per day) and 2 testing days for the training groups: the biomimetic ($n = 20$; mimicking the desired bionic gesture with biological hand) and arbitrary ($n = 20$; mapping an unrelated hand gesture with the desired bionic gesture) groups. After training, we assessed how well the learning for each control strategy would generalize to a new control strategy. We also tested a control group ($n = 20$) that received no bionic hand training (that is, the untrained group). On the basis of

the assumptions underlying biomimetic-inspired design, one would predict that biomimetic control would provide an increased sense of embodiment, better performance, generalization and more intuitive control. By contrast, owing to potential neurocognitive conflict associated with biomimetic control, our (pre-registered) core prediction was that training using an arbitrary control strategy might show increased performance over training and post-training generalization to a new control mapping (for our pre-registration, see https://osf.io/3m592/). Biomimetic control might provide specific advantages in short-term performance and automaticity (more intuitive control). In addition, we predicted that, regardless of the control type, bionic hand skill learning would be associated with an increased sense of embodiment and motor control over the course of training.

## Results

To compare biomimetic and arbitrary control strategies, we tested five key features of bionic hand skill learning: (1) sense of embodiment, (2) early training performance, (3) late training performance, (4) control automaticity and (5) post-training generalization to a new control mapping. To quantify motor performance, we focused on three myoelectric control skills: speed, dexterity and gesture switching.

To ensure any differences in skill learning were not driven by intrinsic differences in motor ability, before training we tested participants on a ballistic reaching task using either the bionic hand (not yet turned on) or their biological left hand. We observed that all three groups had similar pre-training motor ability when wearing the bionic hand ($F_{(2,54)} = 0.009$, $P = 0.991$, eta-squared effect size ($\eta^2$) = $3.1 \times 10^{-4}$, inclusive Bayes factor ($BF_{incl}$) = 0.14) and with their biological hand ($F_{(2,54)} = 1.012$, $P = 0.370$, $\eta^2 = 0.03$, $BF_{incl} = 0.291$; Supplementary Fig. 1). Next, to ensure that any differences in skill learning were not driven by differences between EMG classifier performance for the biomimetic and arbitrary control strategies, we tested classifier performance immediately following device calibration. We found that the classifier had high performance for all participants, with no differences in accuracy between control strategies (biomimetic average classification accuracy, $89 \pm 5\%$; arbitrary, $88 \pm 7\%$; Mann–Whitney statistic ($W$) = 167.0; $P = 0.631$; rank-biserial correlation ($r_{rb}$) = $-0.99$; 95% confidence interval (CI), [$-0.28$, $0.45$]; Bayes factor ($BF_{10}$) = 0.37; Fig. 2). Therefore, any group differences potentially observed in skill learning could not be attributed to intrinsic differences in motor ability or classifier performance between control strategies.

### Both strategies show similar increases in bionic hand embodiment

To compare the two control strategies, we first assessed changes in the perceived (phenomenological) sense of embodiment of the bionic hand. Before and after training, participants were asked to respond to statements related to key embodiment categories: body ownership ('it seems like the robotic hand is part of my body'), agency ('it seems like I am in control of the robotic hand') and visual appearance ('it seems like I am looking directly at my own hand, rather than a robotic hand'; Fig. 3a; for a list of all questionnaire statements see Supplementary Table 1). Comparing pre- with post-training scores, all trained participants reported a significant increase of embodiment in body ownership ($W = 84.0$; $P < 0.001$; $r_{rb} = -0.79$; 95% CI, [$-0.89$, $-0.62$]) and agency ($W = 12.0$; $P < 0.001$; $r_{rb} = -0.96$; 95% CI, [$-0.98$, $-0.93$]), but not visual appearance ($W = 137.0$; $P = 0.132$; $r_{rb} = -0.32$; 95% CI, [$-0.64$, $0.08$]; $BF_{10} = 0.48$; Fig. 3b). As subjective reports are particularly malleable to task demands[17], we also compared the training group with the untrained group (responding to the statements 1 week apart). This allowed us to confirm increased embodiment (post- minus pre-training ratings) in the trained groups relative to the untrained group (for body ownership, $W = 263.0$; $P = 0.045$; $r_{rb} = -0.32$; 95% CI, [$-0.57$, $-0.02$]; for agency, $W = 81.0$; $P < 0.001$; $r_{rb} = -0.79$; 95% CI, [$-0.88$, $-0.64$]; for visual appearance, $W = 169.50$; $P < 0.001$; $r_{rb} = -0.56$; 95% CI, [$-0.74$, $-0.31$];

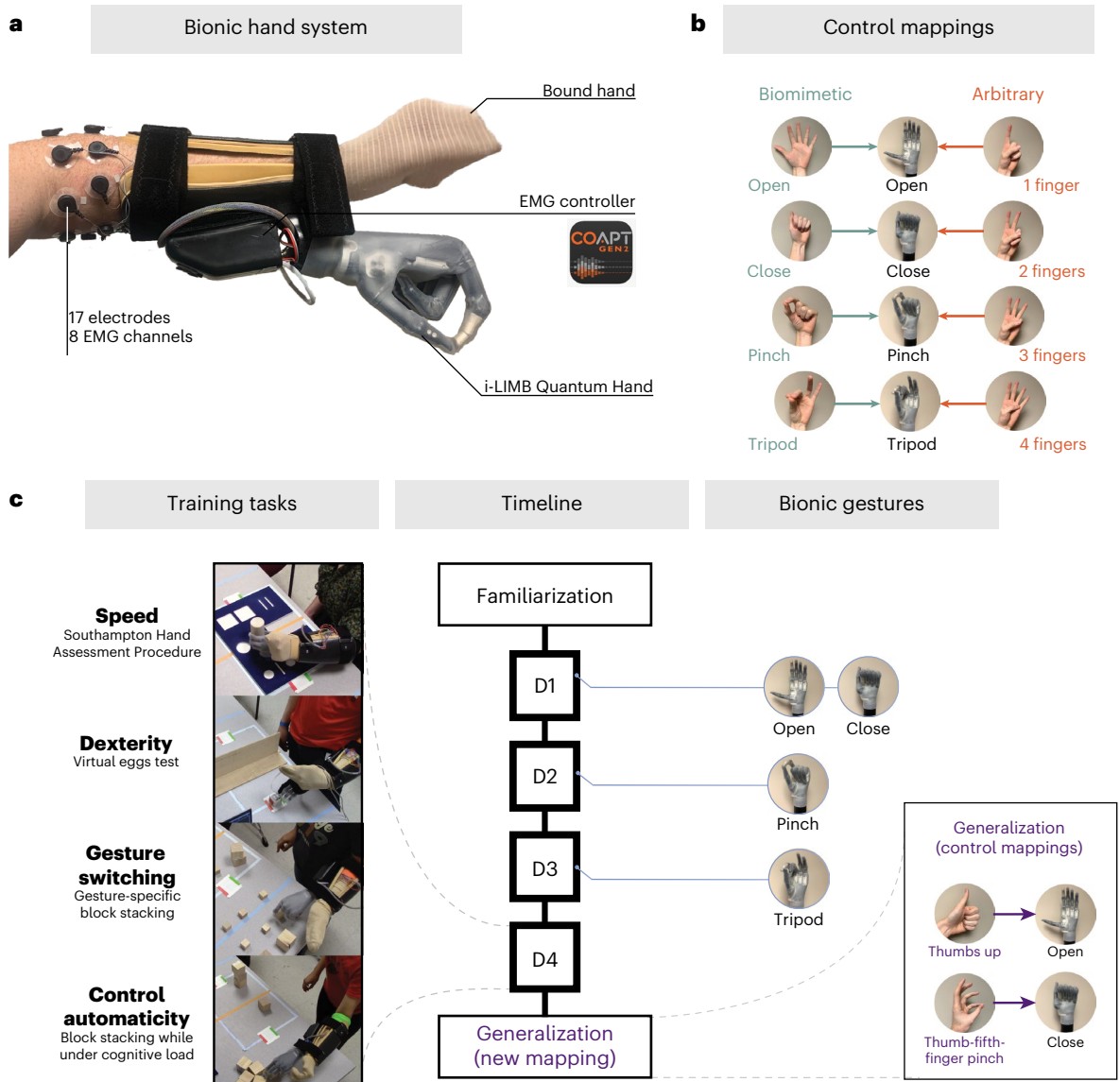

**Fig. 1 | Experimental design of the study. a**, Bionic hand system attached to the participant's left arm. The i-LIMB Quantum bionic hand is controlled by a Coapt pattern-recognition controller (Coapt, Complete Control Gen2) using signals from surface EMG electrodes (eight channels) positioned around the muscles of the forearm (for a detailed breakdown of device components, see Methods). The biological hand was bound to minimize visual differences between the two control strategies. **b**, Biomimetic and arbitrary users calibrated their EMG controller so that specific biological hand gestures would engage specific bionic hand gestures (for the biomimetic strategy, these were matched).

**c**, Experimental design for the trained groups. Left: examples of the training tasks included in a daily training session (Supplementary Video 1). Middle: the timeline for each of the study visits. Right: depicts when the bionic hand gestures were introduced to participants in their training (on D1, open and close; D2, pinch; D3, tripod). In the post-training generalization session, all participants (including the untrained participant group) learnt to control the hand using a new set of hand gestures (that is, new mapping). Coapt Gen2 used with the permission of Coapt LLC.

Fig. 3c). Importantly, we did not find differences between biomimetic and arbitrary users in the magnitude of this increase in embodiment reports, with the biomimetic and arbitrary group showing qualitatively (although not significantly) greater increase for ownership and agency, respectively (for body ownership, $W = 266.50$; $P = 0.143$; $r_{rb} = 0.26$; 95% CI, [−0.08, 0.56]; $BF_{10} = 0.53$; for agency, $W = 193.50$; $P = 0.675$; $r_{rb} = −0.07$; 95% CI, [−0.41, 0.27]; $BF_{10} = 0.35$; note that values are not corrected for multiple comparisons). Overall, in contrast to the common assumptions of biomimetic design, biomimetic control did not provide a statistically significant increased sense of embodiment. Given no differences in perceived embodiment, this raised the interesting question of whether we might observe differences between groups in skill learning.

**Biomimetic control provides some early training speed benefits**
We next focused on early training performance. To measure control speed, we quantified the ability to operate the hand using completion time on a modified version of the Southampton Hand Assessment Procedure (SHAP; Supplementary Video 1). Note that this task had minimal motor requirements, that is, participants were asked to grasp, transport and release the grasp of various objects, and, as such, completion time adequately reflected task performance. During the first training day, all participants were able to successfully complete the task—showing the elementary difficulty level—but biomimetic users were faster than arbitrary users (day 1 (D1) performance, $W = 85.0$; $P = 0.001$; $r_{rb} = −0.59$; 95% CI, [−0.77, −0.31]; Fig. 4a). Next, to quantify dexterity, we used the virtual eggs test, which measures a user's ability to gently grasp and transport

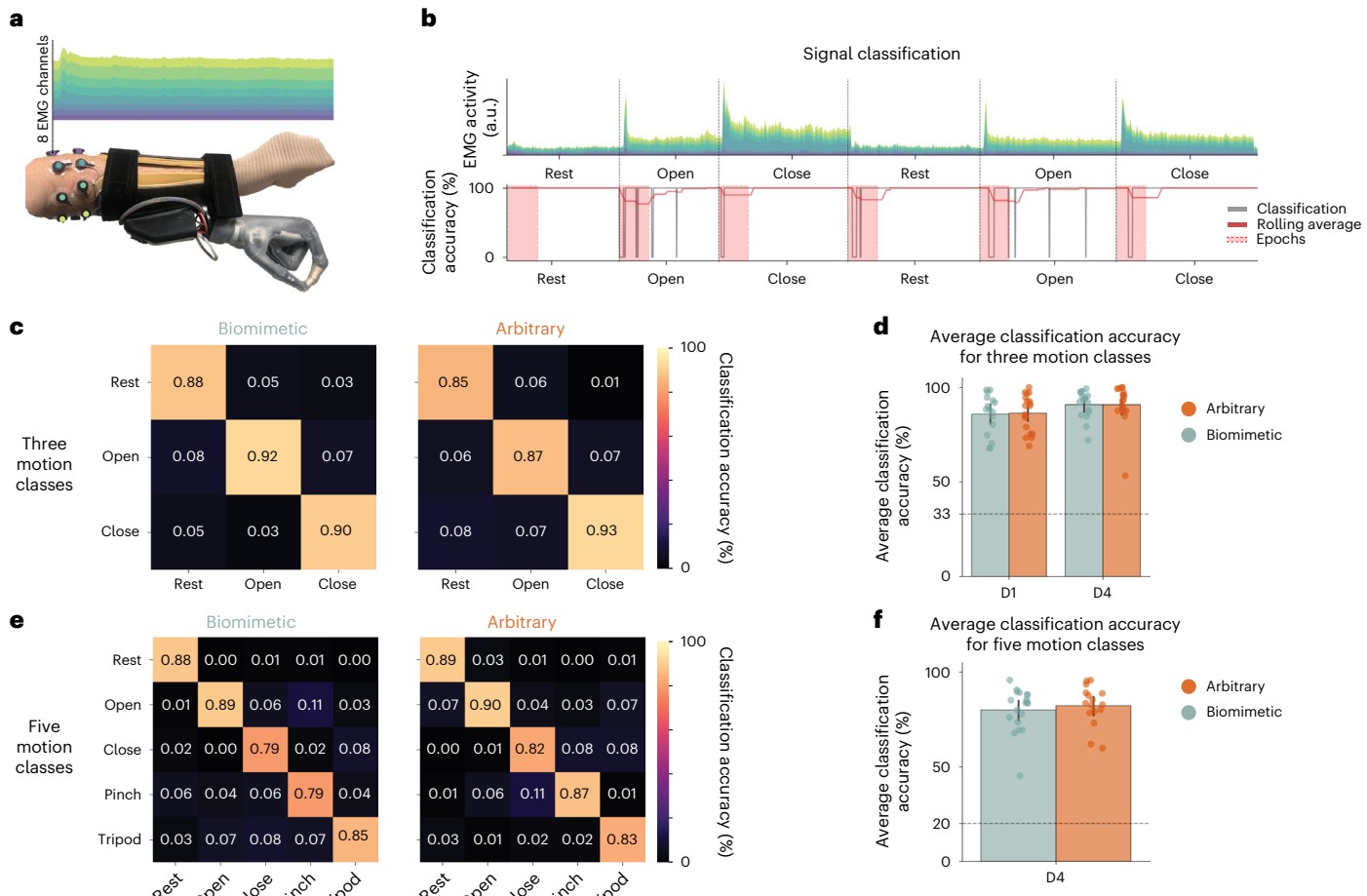

**Fig. 2 | EMG classification accuracy.** EMG data when testing the trained classifier. **a,b**, Example visualizations of the EMG signals acquired from the bionic hand system (**a**) and the real-time EMG signals and classification decisions while an example participant engaged and held each hand gesture for 20 s (**b**) (Methods). During this assessment, the Coapt controller outputted a real-time classification decision (updated every 50 ms) for which gesture was being performed. Comparing this output with the gesture participants were instructed to perform by the experimenter, we computed the classification accuracy for each gesture. The time window of interest was approximately the first 2.5 s of each 20 s gesture trial (shown in **b** as a red window). **c**, Biomimetic ($n = 19$) and arbitrary ($n = 17$) group average classification accuracy matrices for 3 gesture classes (rest, open and close) were assessed on D1, immediately following controller calibration. **d**, Both trained groups ($n = 34$) showed a significant increase in average classification accuracy (main effect of day, $F_{(1,31)} = 5.647$,

$P = 0.024$, $\eta^2 = 2.6 \times 10^{-5}$) between the first (D1) and last day (D4) of training. No significant differences were found between the two training strategies before training ($W = 167.0$; $P = 0.631$; $r_{rb} = 0.09$; 95% CI, [−0.28, 0.45]; $BF_{10} = 0.37$) or after training ($W = 121.50$; $P = 0.437$; $r_{rb} = −0.15$; 95% CI, [−0.50, 0.22]; $BF_{10} = 0.37$). **e**, Biomimetic ($n = 19$) and arbitrary ($n = 16$) average classification accuracy matrices for 5 gesture classes (rest, open, close, pinch and tripod), assessed on D4, at the end of training. **f**, No differences between trained groups ($n = 34$) in average classification accuracy (average of the 5-motion-class diagonal; $W = 117.50$; $P = 0.517$; $r_{rb} = −0.13$; 95% CI, [−0.49, 0.25]; $BF_{10} = 0.42$). The dashed line denotes chance level (3 classes, 33%; 5 classes, 20%). Data plotted in **d** and **f** reflect group means ± s.e.m. All statistical group comparisons were two-tailed Mann–Whitney tests. Circles depict individual participant means (across relevant items). Values indicate group means ± s.e.m.

fragile (magnet fused) blocks (that is, eggs) without breaking them (Supplementary Video 1). This task requires greater motor resources, as shown in previous research[7,18,19]. During the first training day, the majority of participants (75%) could not successfully transfer 1 egg without breaking it within the allocated time—showing the increased difficulty of this task. Importantly, there were no group differences (for D1 performance in number of successful eggs transferred, $W = 185.50$; $P = 0.897$; $r_{rb} = −0.02$; 95% CI, [−0.37, 0.33]; $BF_{10} = 0.34$; for percentage of successful to total eggs transferred, $W = 221.50$; $P = 0.740$; $r_{rb} = 0.05$; 95% CI, [−0.29, 0.39]; $BF_{10} = 0.32$; for the pressure applied by the bionic hand on egg, $W = 205.0$; $P = 0.904$; $r_{rb} = −0.14$; 95% CI, [−0.45, 0.20]; $BF_{10} = 0.33$; Fig. 4b). We also tested the ability of participants to switch between bionic hand gestures. Fluent switching across multiple gestures is considered an advanced ability for prosthesis users. To quantify gesture switching, we used completion time on a block-stacking task that required participants to successfully grasp and transfer objects

using the bionic hand close-and-pinch gestures, switching back and forth (Supplementary Video 1). This ability could only be first tested on D2 because participants were only then trained on the second grasping gesture (pinch), thus providing gesture-switching functionality. During the first attempt of this task, we observed no group differences in performance (for D2 performance, $W = 223.0$; $P = 0.537$; $r_{rb} = 0.11$; 95% CI, [−0.24, 0.44]; $BF_{10} = 0.37$; Fig. 4c). Overall these findings suggest that biomimetic control affords early training benefits in speed, but not in dexterity and gesture switching.

### Biomimetic advantage reduces with more training

Next we examined late training motor performance (that is, across the subsequent training days). For the easier speed task, on the SHAP, biomimetic users continued to outperform arbitrary users. All trained participants continued to improve performance with training (for main effect of day, $F_{(3,114)} = 67.97$, $P < 0.001$, $\eta^2 = 0.32$; Fig. 4a).

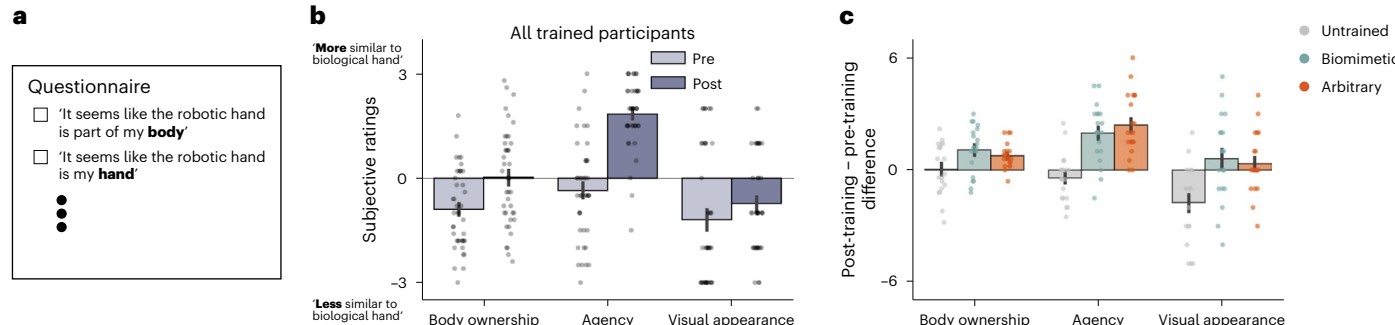

**Fig. 3 | Trained participants show increased sense of bionic hand embodiment, regardless of control strategy. a**, An example item from the questionnaire used, designed to probe subjective sense of embodiment (all statements listed in Supplementary Table 1). Scores were averaged over items of the same category (for example, all body ownership questions averaged into a single value). **b**, After training, trained participants ($n = 41$) showed significant increases in their sense of bionic hand embodiment on statements reflecting

body ownership and agency ($-3$ = disagree, 0 = neutral, $+3$ = agree, or somewhere in between). **c**, Trained participants ($n = 41$) showed significant increases in embodiment scores compared with the scores of untrained participants ($n = 19$) in all categories, regardless of the control strategy with which users trained. There were no significant differences between biomimetic ($n = 21$) and arbitrary ($n = 20$) users. For **b** and **c**, circles depict individual participant means (across relevant items). Values indicate group means ± s.e.m.

Although the group differences narrowed with training (significant interaction between day and group, $F_{(3,114)} = 5.659$, $P = 0.001$, $\eta^2 = 0.03$), biomimetic users were still faster than arbitrary users even in the last day of training (D4; $W = 96.0$, $P = 0.004$, $r_{rb} = -0.51$, 95% CI, [$-0.73$, $-0.20$]). We also tested users on a different version of the SHAP that tests speed when using the pinch and tripod gestures (in subsequent training days, when these gestures were introduced). For these tasks, we observed significant improvements for all trained participants (main effect of day, for pinch, $F_{(2,78)} = 48.435$, $P < 0.001$, $\eta^2 = 0.17$; for tripod, $F_{(1,39)} = 6.517$, $P = 0.015$, $\eta^2 = 0.01$; Fig. 4a), but no group differences on either task (main effect of group, for pinch, $F_{(1,39)} = 1.5 \times 10^{-5}$, $P = 0.997$, $\eta^2 = 2.72 \times 10^{-7}$, BF$_{incl}$ = 0.67; for tripod, $F_{(1,39)} = 0.676$, $P = 0.416$, $\eta^2 = 0.01$, BF$_{incl}$ = 0.38; for all reported comparisons see Supplementary Table 2). This might further reflect the previous observation that group differences on the SHAP speed task were reduced over training.

For the more demanding dexterity task, we did not observe any group differences emerging with training. All trained participants improved in all dexterity measures across the training days (main effect of day, for number of successful egg transfers, $F_{(3,108)} = 12.960$, $P < 0.001$, $\eta^2 = 0.01$; for percentage of total eggs transferred that were successful, $F_{(3,105)} = 8.965$, $P < 0.001$, $\eta^2 = 0.08$; for applied pressure, $F_{(3,102)} = 5.476$, $P = 0.002$, $\eta^2 = 0.03$; Fig. 4b) and there were no differences between groups (main effect of group, for number of successful egg transfers, $F_{(1,36)} = 0.363$, $P = 0.550$, $\eta^2 = 0.005$, BF$_{incl}$ = 0.27; for percentage of total eggs transferred that were successful, $F_{(1,35)} = 0.005$, $P = 0.945$, $\eta^2 = 7.9 \times 10^{-5}$, BF$_{incl}$ = 0.29; for applied pressure, $F_{(1,34)} = 0.090$, $P = 0.765$, $\eta^2 = 0.002$, BF$_{incl}$ = 0.35; for all statistical comparisons see Supplementary Table 2). On the last day of training (D4), both groups performed similarly (for number of successful egg transfers, $W = 209.0$; $P = 0.598$; $r_{rb} = 0.10$; 95% CI, [$-0.26$, $0.43$]; BF$_{10}$ = 0.31; for percentage of total eggs transferred that were successful, $W = 218.0$; $P = 0.631$; $r_{rb} = 0.09$; 95% CI, [$-0.26$, $0.42$]; BF$_{10}$ = 0.32; for applied pressure, $W = 212.0$; $P = 0.550$; $r_{rb} = 0.11$; 95% CI, [$-0.24$, $0.44$]; BF$_{10}$ = 0.35).

Similarly, for the technically challenging gesture switching (that is, switching between the two grasping gestures close and pinch), all trained participants improved in gesture-switching speed (main effect of day, $F_{(2,76)} = 13.766$, $P < 0.001$, $\eta^2 = 0.09$; Fig. 4c) and there were no significant differences in average performance between groups (main effect of group, $F_{(1,38)} = 0.044$, $P = 0.835$, $\eta^2 = 7.0 \times 10^{-4}$, BF$_{incl}$ = 0.63). In addition, when all gestures (3 grasping gestures, close, pinch and tripod) were incorporated into the task, we found no differences between groups ($W = 172.0$; $P = 0.646$; $r_{rb} = -0.09$; 95% CI, [$-0.42$, $0.27$]; BF$_{10}$ = 0.34; Fig. 4d).

Overall we observed a speed advantage for biomimetic users. However, this advantage was seen to reduce with training, and the advantage was only observed for the SHAP close-cylinder object (not the pinch or tripod). In addition, biomimetic control did not show any advantages, relative to the arbitrary strategy, when learning more complex dexterity and gesture-switching control. Instead, we found that training led to improvements, regardless of the control strategy.

**Biomimetic control is more automatic early in training**
Another key component for successful integration with a bionic limb is the ability to multitask, such that attentional resources (for example, focused exclusively on online control and movement planning) can be diverted towards other tasks without interfering with device control (that is, control automaticity). On the first and last days of training, we tested the impact increased cognitive load would have on motor performance. We implemented a dual cognitive–motor task that required participants to perform arithmetic operations while simultaneously using the bionic hand to stack blocks (Supplementary Video 1). To quantify the impact of increased cognitive load, we compared the number of blocks stacked with the counting task versus without (baseline). For counting performance, we observed that both groups performed the task similarly (main effect of group, $F_{(1,38)} = 0.025$, $P = 0.874$, $\eta^2 = 4.3 \times 10^{-4}$, BF$_{incl}$ = 0.32). For motor performance, we observed significant differences between groups across days (interaction between day and group, $F_{(1,38)} = 9.896$, $P = 0.003$, $\eta^2 = 0.06$; Fig. 5a). Looking at the first and last days separately, we observed that on the first day of training, arbitrary users were more affected by the cognitive task than biomimetic users ($W = 286.50$; $P = 0.019$; $r_{rb} = 0.43$; 95% CI, [$0.10$, $0.68$]). However, on the last day of training, both groups were similarly affected by the cognitive load ($W = 176.0$; $P = 0.533$; $r_{rb} = -0.11$; 95% CI, [$-0.44$, $0.24$]; BF$_{10}$ = 0.36), suggesting that biomimetic control is more automatic early in training compared with arbitrary control, but arbitrary control becomes just as intuitive with continued training.

Another means of measuring automaticity is by simply asking participants to rate their subjective sense of control difficulty at the end of every training day (Methods). In these reports, all trained participants reported a significant decrease in control difficulty across days (main effect of day, $F_{(3,114)} = 21.298$, $P < 0.001$, $\eta^2 = 0.16$; Fig. 5b), but there were no average group differences in ratings across days (main effect of group, $F_{(1,38)} = 0.041$, $P = 0.840$, $\eta^2 = 5.7 \times 10^{-4}$, BF$_{incl}$ = 0.56).

Overall we observed that biomimetic control provided increased control automaticity early in training compared with arbitrary control. However, this advantage diminished when assessed later in training.

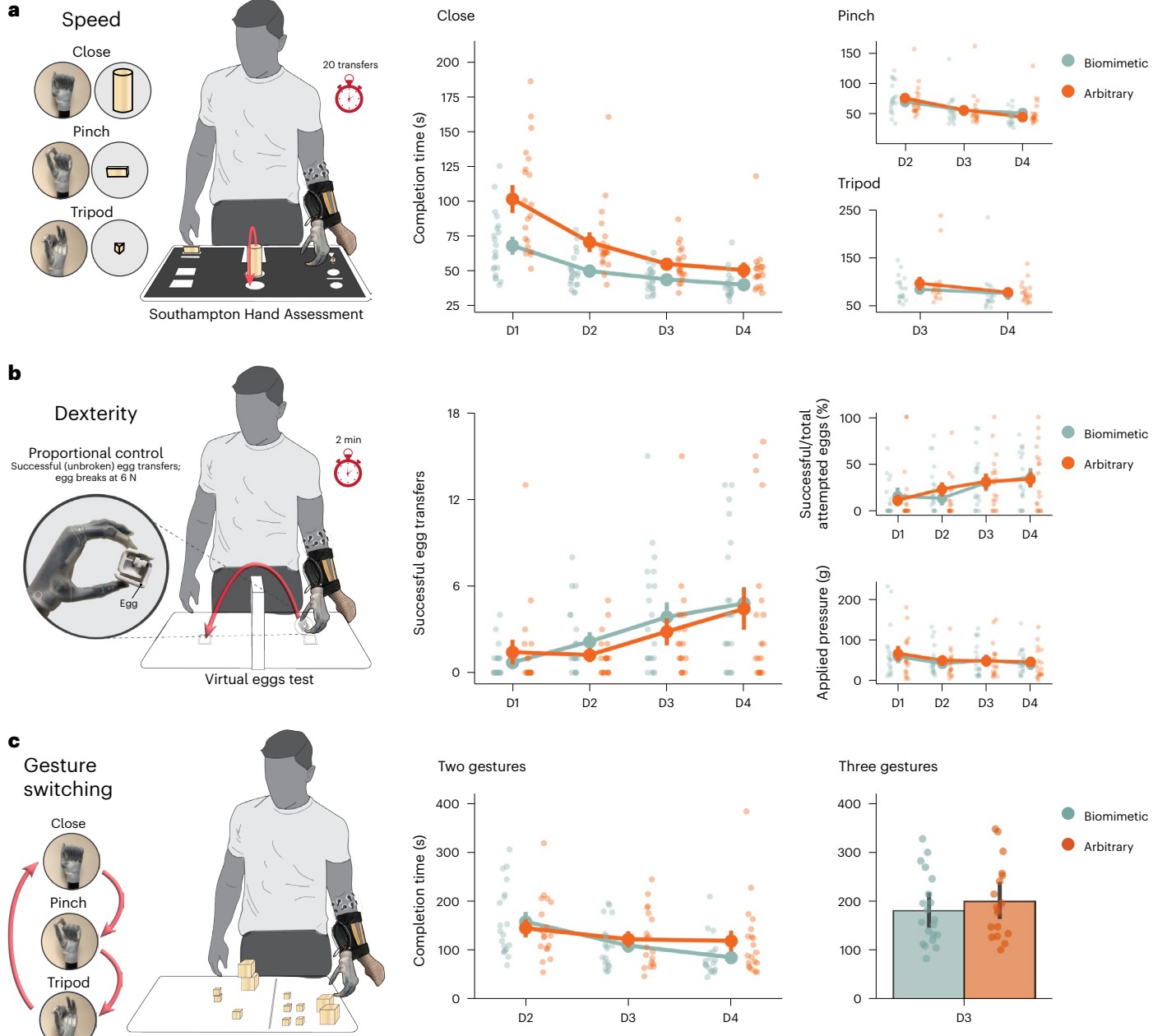

**Fig. 4 | Bionic hand skill learning on speed, dexterity and gesture-switching tasks. a**, Trained participants (*n* = 41) improved control speed on all bionic gestures. For the close gesture, biomimetic control was faster than arbitrary control across training sessions. **b**,**c**, Trained participants (*n* = 41) improved in control dexterity (number of successful or unbroken egg transfers, percentage of total attempted eggs transferred that were successful, and pressure applied by thumb and index digits during task) (**b**) and gesture switching (**c**) across training sessions, regardless of control strategy. For gesture switching, because participants were trained on new grasping gestures each training day, we used two versions of this task. The two-gestures version required successfully switching between close and pinch bionic gestures. The three-gestures version required successfully switching between close, pinch and tripod bionic gestures. No significant differences were found between control strategies. See Supplementary Video 1 for examples of all tasks. Circles depict individual participant means (across relevant items). Values indicate group means ± s.e.m. All other annotations are the same as described in Fig. 2.

In addition, we did not find a significant impact of the control strategy on the subjective experience of control difficulty.

**Initial biomimetic gestures have more separable EMG patterns**

What could be driving the speed delay for arbitrary users early in their training? The most obvious explanation is the cognitive disadvantage for arbitrary users (for example, the time it takes to regenerate the relevant movement to actuate the grasp), as seen in the dual cognitive–motor task findings. Alternatively, there could be differences in the EMG gestural structure of the control strategies (for example, some gestures could be more difficult to precisely articulate, relative to other gestures). In other words, although we observed similar EMG classification accuracy across groups (Fig. 6), there might still be a greater cost on one group to achieve the same level of classification accuracy. To explore this, we investigated the EMG data participants generated when training their classifier. We visualized the EMG profiles across channels at the group and individual participant level (Fig. 6a). To quantify the similarity between gestures across EMG channels, we computed the euclidean distance between all gesture combinations (Fig. 6b). From this analysis, we observed that across all trained gestures there were

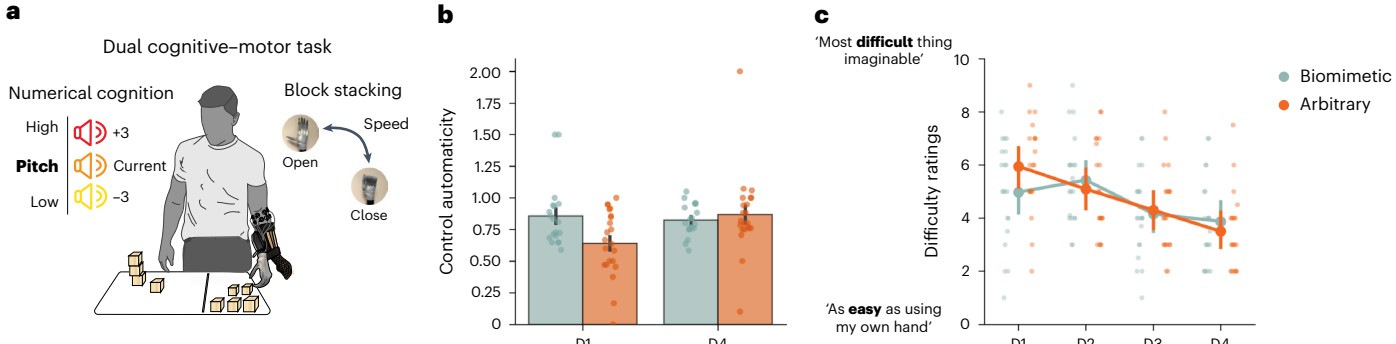

**Fig. 5 | Arbitrary control is less automatic early in training but similar to biomimetic later. a**, Schematic of the dual cognitive–motor task requiring users to simultaneously perform a cognitively demanding arithmetic task aloud (left) while performing a block-stacking task (right). Control automaticity was computed by dividing motor performance (number of blocks stacked) when simultaneously performing a counting task by motor performance alone. By this measure, higher values indicate less cognitive encumbrance, or more automatic control. **b**, Biomimetic control showed superior performance earlier in training compared with arbitrary control. However, by the end of training, arbitrary control produced similar performance to that observed in the biomimetic group. **c**, All trained participants reported control got easier across the training sessions, regardless of the control strategy. For **b** and **c**, circles depict individual participant means (across relevant items). Values indicate group means ± s.e.m. All other annotations are the same as described in Fig. 2.

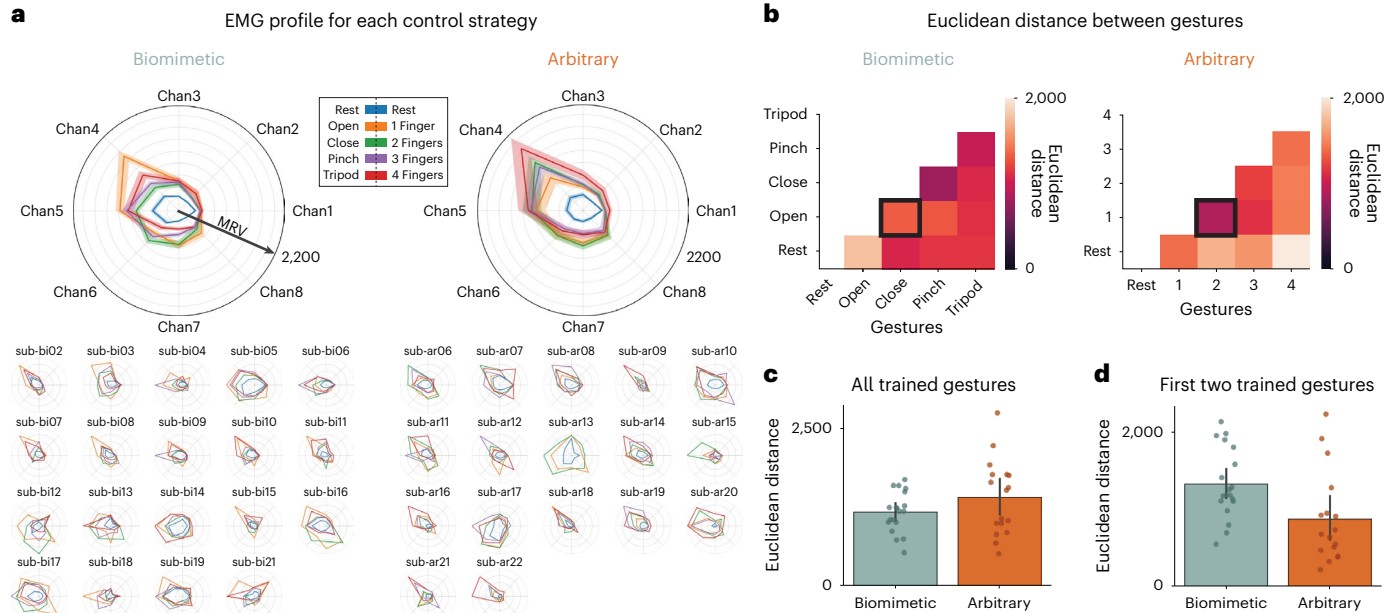

**Fig. 6 | EMG training data.** EMG data when training the classifier. **a**, A visualization of the EMG MRV profile for the biomimetic ($n = 19$) and arbitrary ($n = 17$) group average (top) and individual participants (bottom) for each gesture across the 8 surface EMG channels (Chan1–Chan8) around the forearm (using the EMG data participants generated to train the classifier). Values reflect the average EMG amplitude at each channel when holding the gesture during calibration. **b**, A matrix filled with the euclidean distances between each gesture pair across channels. **c**, No differences between trained groups in average euclidean distance across all trained gestures ($W = 131.0$; $P = 0.346$; $r_{rb} = -0.18$; 95% CI, [−0.51, 0.19]; $BF_{10} = 0.73$). **d**, A group comparison between EMG profiles for their first 2 trained gestures between the arbitrary (1 finger and 2 fingers; $n = 17$) and biomimetic (open and close; $n = 19$) users revealed more distinct patterns in the biomimetic group ($W = 250.0$; $P = 0.004$; $r_{rb} = 0.54$; 95% CI, [0.22, 0.76]). Data plotted in **c** and **d** reflect group means ± s.e.m. All statistical group comparisons were two-tailed Mann–Whitney tests. Circles depict individual participant means (across relevant items). Values indicate group means ± s.e.m.

no significant group differences ($W = 131.0$; $P = 0.346$; $r_{rb} = -0.18$; 95% CI, [−0.51, 0.19]; $BF_{10} = 0.73$; Fig. 6c). However, for the first two trained gestures (open and close) used on D1, arbitrary users have significantly smaller euclidean distances, that is, more similar distances, relative to biomimetic users ($W = 250.0$; $P = 0.004$; $r_{rb} = 0.54$; 95% CI, [0.22 0.76]; Fig. 6d). In other words, arbitrary users would need to generate more precise muscle contractions to achieve the same level of classification accuracy as biomimetic users (at least for the first two trained gestures).

To further explore whether control automaticity or euclidean distance between gestures drives the group differences we observed on the speed task, we added each measure as a regressor and tested for

group differences in the unstandardized residuals of SHAP D1 scores. When controlling for euclidean distance, we still observed a significant group difference in speed ($t(33) = -2.375$; $P = 0.024$; Cohen's $d = -0.80$; 95% CI, [−1.43, −0.10]) and, when controlling for control automaticity, we observed no significant group difference ($t(37) = -1.912$; $P = 0.064$; $d = -0.61$; 95% CI, [−1.25, 0.03]; $BF_{10} = 1.281$). These analyses indicate that the group differences in speed observed on D1 may be exacerbated by a combination of the EMG gestural similarity of the control strategies and the control automaticity surrounding the strategy mapping, with the latter perhaps playing a greater role (see Supplementary Fig. 2 for correlations between measures).

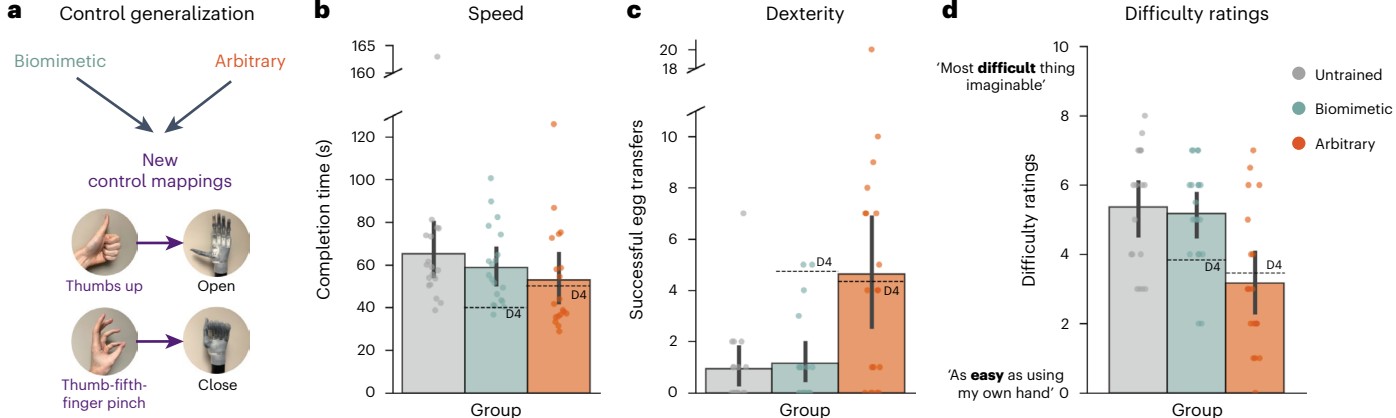

**Fig. 7 | Arbitrary users show increased generalization. a,** During the final study session, all users swapped to the same new control mapping strategy (thumbs up = open hand; thumb–fifth-finger pinch = close hand). **b**–**d**, Performance on speed (**b**), dexterity (**c**) and control difficulty (**d**) assessments for the generalization session. As a reference for previous performance, the dashed line denotes group mean performance on the last (D4) training session with their original control mappings. Biomimetic users (*n* = 20) showed significant impairments compared with performance during D4. Furthermore, performance

in the generalization tasks was similar between the biomimetic group (*n* = 20) and untrained participants (*n* = 19) using the bionic hand for the first time. Arbitrary users (*n* = 18) showed increased generalization such that performance with the new control strategy in the generalization session was similar to performance with the trained strategy on D4. All other annotations are the same as described in Fig. 2. For **b**–**d**, circles depict individual participant means (across relevant items) and values indicate group means ± s.e.m.

## Arbitrary strategy increases generalization to new control mappings

The ability to perform under a different set of conditions is a crucial aspect of prosthesis control. To test how well the learning for each control strategy generalizes to a new control mapping, we included a final post-training testing session where we re-calibrated the controller for all participants with a new set of gestures (Methods and Fig. 7a). All participants used the same set of gestures for this session. With this new control mapping, we tested users' speed, dexterity and control difficulty. We also trained the untrained users to operate the hand with these gestures, providing a baseline for first-time use.

First, when testing speed on the SHAP in the generalization session, we found the biomimetic group matched performance of the untrained users ($W = 202.0$; $P = 0.534$; $r_{rb} = 0.12$; 95% CI, [−0.24, 0.45]; $BF_{10} = 0.38$; Fig. 7b). By contrast, the arbitrary group was significantly faster than untrained users ($W = 239.0$; $P = 0.039$; $r_{rb} = 0.39$; 95% CI, [0.04, 0.66]). For trained participants, we also compared generalization speed performance to their last (D4) training day performance when using their original control mappings. Specifically, we found that biomimetic users' speed dropped significantly between the 2 days (D4 versus generalization, $W = 10.0$; $P < 0.001$; $r_{rb} = −0.90$; 95% CI, [−0.96, −0.76]), whereas arbitrary users' speed did not change (D4 versus generalization, $W = 97.0$; $P = 0.953$; $r_{rb} = 0.21$; 95% CI, [−0.45, 0.48]; $BF_{10} = 0.23$).

Second, when testing dexterity in the generalization session, we similarly observed that biomimetic user performance matched the untrained group ($W = 170.0$; $P = 0.761$; $r_{rb} = −0.05$; 95% CI, [−0.40, 0.30]; $BF_{10} = 0.33$; Fig. 7c), even returning to their pre-training (D1) performance level ($W = 17.50$; $P = 0.328$; $r_{rb} = −0.36$; 95% CI, [−0.79, 0.30]; $BF_{10} = 0.35$). By contrast, the arbitrary group performed significantly better than both the biomimetic ($W = 104.50$; $P = 0.013$; $r_{rb} = −0.45$; 95% CI, [−0.69, −0.11]) and untrained ($W = 88.50$; $P = 0.010$; $r_{rb} = −0.48$; 95% CI, [−0.71, −0.14]) groups. When comparing performance across days directly (D4 versus generalization), we observed significant group differences in dexterity across days (interaction between day and group, $F_{(1,37)} = 6.603$, $P = 0.014$, $\eta^2 = 0.04$). Specifically, we found that biomimetic users' dexterity dropped significantly between the 2 days ($W = 116.0$; $P = 0.002$; $r_{rb} = 0.93$; 95% CI, [0.80, 0.97]). By contrast, arbitrary users' dexterity was the same as that on their previous training

day ($W = 39.50$; $P = 0.70$; $r_{rb} = −0.13$; 95% CI, [−0.63, 0.44]; $BF_{10} = 0.24$), even performing slightly better qualitatively (D4 average eggs, 4.4; generalization average eggs, 4.6). In other words, whereas the biomimetic group showed no indication of generalization from the original to the new control mapping, the arbitrary strategy led to a full generalization of learning.

Finally, we observed that users' subjective sense of control difficulty in the generalization session also supported these findings. We found that biomimetic users rated control difficulty in the generalization session similarly to the untrained group learning to operate the device for the very first time ($W = 194.0$; $P = 0.688$; $r_{rb} = 0.07$; 95% CI, [−0.28, 0.42]; $BF_{10} = 0.33$; Fig. 7d), and even as difficult as biomimetic users' first training day (D1 versus generalization, $W = 72.50$; $P = 0.867$; $r_{rb} = −0.05$; 95% CI, [−0.53, 0.45]; $BF_{10} = 0.25$). By contrast, arbitrary users reported control to be significantly easier than did both the biomimetic ($W = 310.0$; $P = 0.003$; $r_{rb} = 0.55$; 95% CI, [0.25, 0.75]) and untrained ($W = 287.50$; $P = 0.002$; $r_{rb} = 0.59$; 95% CI, [0.30, 0.78]) groups. Furthermore, arbitrary users reported the difficulty to be as easy as their previous training day with their original control mapping ($W = 104.0$; $P = 0.427$; $r_{rb} = 0.21$; 95% CI, [−0.29, 0.63]; $BF_{10} = 0.29$) and qualitatively even slightly easier (D4 mean, 3.58 out of 10; generalization mean, 3.07 out of 10). When comparing responses across days directly (D4 versus generalization), we observed significant differences in ratings between biomimetic and arbitrary groups (significant interaction between day and group, $F_{(1,38)} = 6.857$, $P = 0.013$, $\eta^2 = 0.05$). Specifically, we found that biomimetic user ratings were significantly more difficult in the generalization session compared with D4 ($W = 15.50$; $P = 0.012$; $r_{rb} = −0.74$; 95% CI, [−0.91, −0.36]). By contrast, arbitrary user ratings were equally as difficult as on their previous training day ($W = 104.0$; $P = 0.427$; $r_{rb} = 0.21$; 95% CI, [−0.29, 0.63]; $BF_{10} = 0.27$), qualitatively even slightly easier (D4 rating, 3.5; generalization rating, 3.1). Collectively, this shows that arbitrary training provides increased control generalization to a new control mapping.

## Discussion
It is a widely held assumption that control strategies designed to mimic the biological body might provide unique benefits to the user in terms of device learning, generalization, sense of embodiment and automaticity[4,7,10,20–27]. By contrast to this view, using a wearable EMG-controlled

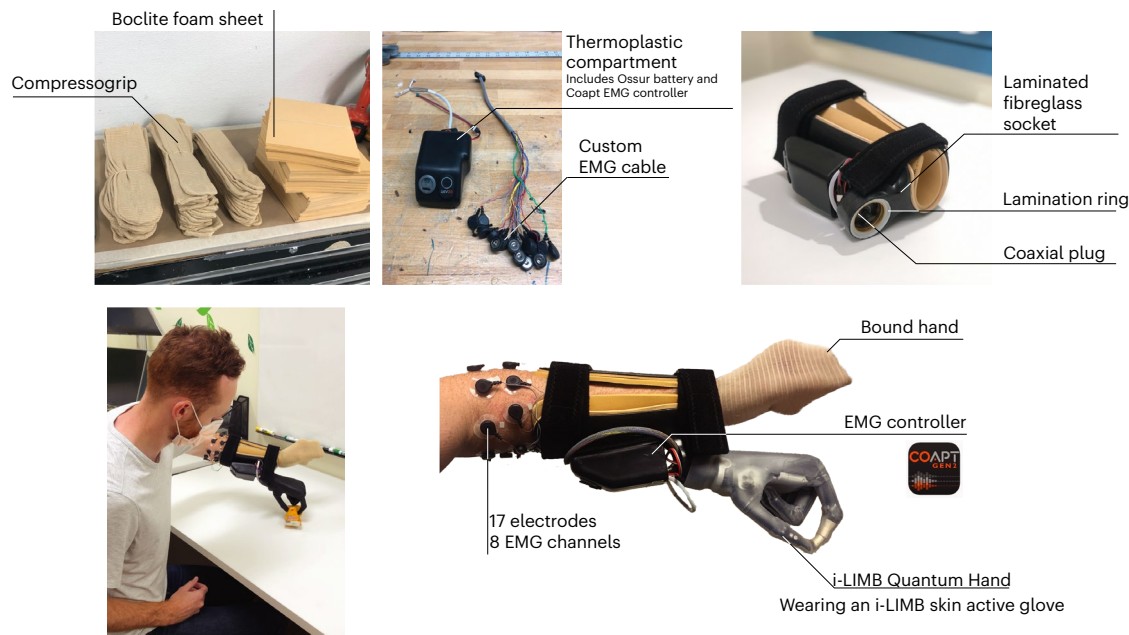

**Fig. 8 | Bionic hand device components.** An image breakdown of all the individual components incorporated in the bionic hand system. Coapt Gen2 logo used with the permission of Coapt LLC.

bionic hand (Fig. 8), across a multitude of tasks, we observed few advantages for biomimetic control. We confirmed our predictions that biomimetic control was more intuitive for users, particularly in the early stages of learning (based on the cognitive load and speed tasks). Subsequent analyses found that these early speed differences were probably driven by a combination of cognitive and EMG factors. However, when task difficulty was increased (more complex gestures, dexterous grasping and gesture switching), these advantages were mostly absent when compared with arbitrary users. Furthermore, we observed that arbitrary users showed increased generalization to a new control mapping, whereas biomimetic users showed less capacity to generalize, performing similarly to untrained participants. In addition, users' subjective experience of perceived embodiment and control difficulty was not impacted by control strategy. Collectively, our findings provide a more balanced perspective of the neurocognitive challenges and opportunities of biomimetic and non-biomimetic control strategies. By challenging some of the core assumptions underlying biomimetic-inspired design, our findings open up the potential for non-biomimetic control solutions for users.

Our findings are consistent with anecdotal evidence suggesting that arbitrary control approaches can be viable and even advantageous for artificial limb control. For example, the previous prosthetic hand winners at the Cybathalon—the Olympics for bionic technology—used devices that were explicitly designed to maximize functionality over biomimicry (2016 winner, Bob Radoc (pilot), DIPO Power Team (designers), Grip 5 Prehensor hand (device); 2020 winner, Krunoslav Mihic (pilot), Andrj Đukić (designer), Maker Hand (device)[28,29]). In addition, on virtual tasks, research in amputees has shown that arbitrary myoelectric control in these individuals can be learnt[30], with one study showing amputees can learn to control an eight-target arbitrary myoelectric interface[31,32]. Other studies have highlighted the emergence of arbitrary muscle synergies for EMG control[33,34]. Most recently, new evidence supporting the versatility of non-biomimetic control has come from the field of motor augmentation. For example, people are able to rapidly learn to use a new body part (an extra robotic thumb) using 2 degrees of freedom operated via the users' toes[35,36]. Finally, efforts to prioritize functionality over biomimicry has led to the development of multiple creative and compelling control schemes for prosthetics (for example, the use of foot switches, body-powered devices, harnessing linear potentiometers, inertial measurement units, radio-frequency identification tags, proximity switches and co-contraction switches). Collectively, this evidence highlights the immense promise of non-biomimetic control solutions for assistive bionics.

Why did arbitrary control provide a comparable degree of skill, and even outperform biomimetic control on generalization? Paradoxically, this could be a result of the cognitively challenging task demands related to adopting atypical gestures for grasp control. Although this conflict would appear to be disadvantageous early in training, there is ample evidence to suggest that this could actually be the key to improve generalization. The first potential explanation has to do with the increased contextual novelty involved with arbitrary training. Consider how our ability to skilfully use our hands is developed over a lifetime of experience, in which information on motor synergies, object knowledge and action semantics is meticulously weaved to construct efficient motor command pipelines[37]. Arbitrary users have to learn to associate hand motor commands with entirely unrelated action goals (previously associated with different motor commands). Research has shown that learning with a more varied input that increases task difficulty is initially slower, but typically yields better generalization (for a review see ref. 38). Similarly, stimuli that incorporate a broader range of neural and cognitive processes have been shown to promote increased generalization of learning[39]. Although speculative, it is possible that the arbitrary learning promotes users to develop more general, adaptable representations of action. A second, related, potential explanation is that arbitrary training requires more cognitive and attentional resources. Research has shown that more complex associations end up being learnt more robustly and are better retained than easy-to-learn associations[40]. Easy-to-learn associations (in our study, biomimetic control) have been shown to rely more heavily on working memory, which allows for very fast learning of information that is not durably retained, whereas learning complex associations, requiring more attention, relies on reinforcement learning that allows for slow, integrative learning of associations that is robustly stored[40,41]. Combined with our observation that the arbitrary group had to work with more similar EMG patterns when first learning to use the bionic

hand, the increased cognitive resources required for arbitrary learning may potentially explain why, when similar learning was achieved by both groups, the arbitrary control outperformed biomimetic control when generalizing to a new control mapping.

The considerations above suggest why arbitrary strategies can be beneficial, but we must also consider the alternative perspective—that is, could biomimetic control be disadvantageous? There are several potential explanations for why biomimicry could have disadvantages, primarily due to the ambition to stay so close to the body. Whereas modern bionic limbs are increasingly biomimetic in design and control, they are not at all the same as biological limbs. Modern prosthetics have speed delays and limited dexterity and functionality, making their operation a lot clumsier than a biological hand. Devices are generally heavy, not particularly durable and their sensory feedback is impoverished. These discrepancies between biological and bionic limb control may be promoting an 'uncanny valley'-type phenomenon for users[42]. The uncanny valley hypothesis proposes that an individual's response to a humanoid robot shifts from empathy to repulsion as the humanoid approaches, but fails to achieve, humanlike appearance (for example, Tom Hanks in the *Polar Express* and the humanoid horror doll in *M3gan*). Although this phenomenon is often used in relation to how we see artificial bodies, the framework also seems relevant to how we control artificial bodies. As device control becomes more and more biomimetic, but not capable of reaching true biomimicry, it may be creating a more active neurocognitive competition between the priors, sensory predictions and motor commands for how amputees controlled their pre-existing limb and how they represent and control their prosthetic limb. Therefore, non-biomimetic control solutions may be more beneficial because the motor control plan can be developed from scratch (that is, independent of pre-existing biological limb control) and can, therefore, avoid any potential conflict.

One could ask: is it at all important to generalize across control strategies for real-world myoelectric prosthetic use? Consider that in a recent survey of over 400 users of prosthetics, 42% of users reported actively using two or more terminal prosthetic devices[43]. Each of these devices—be it a cosmetic, functional hook, simplistic myoelectric or multi-articulating myoelectric—requires entirely different control principles to operate. As such, for users today, an essential component of successful prosthetic use is having flexible, adaptable control or the ability to constantly swap, plug-in and play with a variety of terminal devices without hindering performance. Beyond completely altering control strategies to a new terminal device, control generalization is also an essential requirement for users to operate even a single myoelectric control strategy for a single terminal device due to external and internal factors. Externally, consider that the myoelectric interface—between the surface EMG electrodes and the residual limb—is highly inconsistent throughout the day[44] due to issues pertaining to socket fit, which can often be exacerbated by sweat build-up and/or rain or humidity. Internally, changes in arm posture during muscle contraction have been shown to impact muscle geometry in different ways (muscle fibre length, diameter and orientation), which in turn can systematically alter the EMG features for a specific muscle[45,46]. Consequently, a sensor can receive highly variable signals for the same intended contraction. Therefore, users must have flexible, generalizable representations of control gestures, such that their control and motor learning are not strictly dependent on a single strategy, arm posture, time of day, terminal device, task, and so on. This is why the increased control generalization for the arbitrary group is so important for how we think about real-world prosthetic use and the future of biomimetic prosthetics.

It is important to note that our experimental design may have several potential limitations. First, due to the limited state of modern prosthetic technology (for example, delay between execution of the motor command to fully engaging motors into the actuated grasp), the biomimetic strategy was not purely biomimetic (that is, the same as

biological hand control). In addition, even in the relatively rich realm of pattern recognition, the number of different patterns that are available for control tend to be limited, that is, restricted to discrete and sequential classification. This means that the natural control that we envisage when considering biomimetic strategies is currently reduced to a limited set of movements. It is possible that as technology progresses to resolve some of these basic bottlenecks, the biomimetic approach will provide more immediate translation. However, by the same token, and given the evidence provided here, it is also possible that as the gap between natural and bionic movement narrows, the conflict of the increasingly more subtle differences in motor demands grows. Regardless of these limitations in modern technology, if we consider biomimicry as a spectrum of strategies closer and further away from the biological body, the biomimetic control we tested falls closer to biological control than arbitrary control. However, this raises an important consideration for the scalability of arbitrary control in future bionics; that is, as the number of arbitrary mappings increases, is the cognitive load to learn and consolidate these mappings too demanding for a user? Although this may not be the case, a relevant example to consider is the popularity of video games. Most modern video game controllers have high control dimensionality to enable the precise control of a virtual avatar or effector. Yet millions, even billions, of individuals successfully learn and master the large number of arbitrary mappings between controller and virtual effector with ease. A second limitation is that our training was restricted to four daily laboratory-based sessions, and it is possible that with additional training more differences would have emerged between the groups. Although this might seem like limited training, a recent survey reported that about half (43%) of amputees that were trained to use a prosthesis received between just 1 and 3 training visits with their device[15]. Furthermore, the arbitrary gestures we tested involved simple finger extensions, and it is likely that more careful curation of control mapping could have produced much greater benefits for arbitrary control. Indeed, considering the first two trained arbitrary gestures showed more similar EMG patterns than biomimetic gestures, a key methodological strength of the arbitrary approach is the freedom to design optimal arbitrary mapping strategies with more distinguishable classification accuracies to biomimetic strategies. For these potential limitations, in our view, the fact that arbitrary users matched biomimetic performance after such minimal training is promising. A third limitation is that perhaps arbitrary users showed greater generalization than biomimetic users simply because the generalization session control mappings are essentially just a different arbitrary strategy. However, regardless of why the arbitrary users showed an increased generalization, why did the biomimetic users show no advantage relative to first-time users in the generalization session? Consider that during the generalization session, the no-training control group (users that have never used the device before) perform the same as the biomimetic group in speed ($BF_{10} = 0.38$), dexterity ($BF_{10} = 0.33$) and subjective sense of control difficulty ($BF_{10} = 0.33$). Despite the biomimetic group spending 4 days (approximately ≥8 h) training to operate the device under a variety of conditions and tasks, altering their control strategy completely annihilated their motor performance, such that it is as if they had never trained at all. Although we cannot disregard the potential bias the choice of generalization gestures had for arbitrary users, the fact that biomimetic skill learning was so highly constrained to their control mapping is what is so interesting. A final limitation is that we tested non-disabled participants using the bionic hand to replace their existing hand instead of participants who were prosthesis users, and who might incur different benefits and costs to the biomimetic approach. However, considering the primary aim of the study was to mimic bionic limb control to the biological body, we thought it was necessary for us to have direct access to participants' biological limbs to ensure true biomimicry. Moreover, research is now mounting to indicate that amputation does not induce far-reaching changes to the motor representation of the missing limb[47–49], even

with regards to motor learning[50]. Regardless, future studies should continue to evaluate personalized control solutions for individuals with a missing limb.

In summary, owing to the current limitations in modern prosthetic technologies not yet matching those available to Luke Skywalker, researchers and engineers should continue exploring both biomimetic and non-biomimetic control solutions. Although biomimetic design is an understandable starting point when designing human–machine interfaces, it should not necessarily be the ultimate, end-all goal. From a cognitive neuroscience perspective, there are multiple considerations for and against biomimetic control. So, given the modern-day technological context of biomimetic control, practically speaking, how biomimetic should we go? On the basis of our findings, this depends entirely on the purpose of the prosthesis. If the device is intended for short-term use, simple functionality or where training opportunities are limited, then biomimetic-inspired control options make a lot of sense. However, if the purpose is to design versatile devices with multiple functions for long-term use, at least with modern EMG pattern-recognition technology, non-biomimetic control solutions may provide a useful means to enhance certain aspects of bionic hand motor learning. Furthermore, abandoning the unrealistic ambition of true biomimicry opens up endless possibilities for users and engineers to develop a variety of different control solutions. We suggest that engineers and prosthetists involved in the commercial and clinical delivery of this technology should prioritize flexibility—educating users on the spectrum of biomimetic-to-arbitrary control strategies available to them such that personalized, user-specific control strategies can be selected based on individual user requirements. In our experience, when users are educated with the knowledge and confidence that multiple control approaches are possible, there is a higher likelihood that devices will meet user expectations and requirements. Personalized control strategies will help to propel the industry closer to the actual goal: more satisfied prosthesis users.

## Methods

The study and its experimental procedures were approved by the National Institutes of Health (NIH) Institutional Review Board (NCT00001360, 93M-0170). This study was conducted as part of a larger experiment (see https://osf.io/3m592/ for pre-registration of the full experimental protocol). Below we only report the methodology relevant for the results detailed in this Article.

### Participants

Sixty-one healthy volunteers (40 women; mean age, $24.8 \pm 0.66$ years; all right handed) were recruited from the NIH community and the Washington, DC metro area and were randomly assigned to 1 of the following study groups: biomimetic ($n = 21$; 14 women; mean age, $23.9 \pm 0.57$ years), arbitrary ($n = 21$; 12 women; mean age, $25.9 \pm 1.28$ years) or untrained ($n = 19$; 14 women; mean age, $24.6 \pm 1.41$ years). All participants were unaware of the other participant groups to minimize any potential biases on participant performance. All participants had no known motor disorders. Criteria for participant inclusion were determined before data collection according to the study population guidelines approved by the NIH Institutional Review Board as a part of the study protocol (93-M-0170, NCT00001360). All participants gave their written informed consent before participating in the study and were compensated monetarily for their time. In addition, research participants provided written informed consent for publication of the images and videos depicted in Fig. 1 and Supplementary Video 1.

Two additional participants were recruited, but not included in this study due to incomplete datasets.

### Experimental design

To quantify bionic hand skill learning, we implemented a longitudinal experimental design (Fig. 1c) involving 6 experimental sessions conducted across 6 days (1 session per day, within a 1 week period), as summarized in Fig. 1. All trained participants (biomimetic and arbitrary groups) underwent (1) a familiarization session (2 h), introducing the equipment and completing some pre-training motor control assessments; (2) 4 training sessions (2–3 h) conducted over 4 consecutive days (1 session per day); and (3) a final generalization behavioural session (2 h).

Untrained participants underwent a modified schedule: (1) the familiarization session (2 h) and (2) the generalization behavioural session (1 week later, 2 h). The generalization session was the first time untrained participants were able to have active control over the device.

### Biomimetic, arbitrary and generalization control mappings

Biomimetic and arbitrary control mappings differed in the biological hand gestures that were required to engage the bionic hand classifier (Fig. 1b). The biomimetic control mappings included: open hand (biological) = open hand (bionic), close hand (biological) = close hand (bionic), pinch (biological) = pinch (bionic) and tripod (biological) = tripod (bionic). The arbitrary control mappings included: one finger (biological) = open hand (bionic), two fingers (biological) = close hand (bionic), three fingers (biological) = pinch (bionic) and four fingers (biological) = tripod (bionic). Finally, during the generalization session, all participants (untrained participants included) learnt a new control strategy. The generalization control mappings included: thumbs up (biological) = open hand (bionic), and thumb–fifth-finger pinch (biological) = close hand (bionic). The generalization gestures were intended to be gestures that were a hybrid of the gestures in the biomimetic and arbitrary gestures. In addition, our reasoning for having all groups use the same gestures in this session was that it would allow us to directly match performance across groups.

### Bionic hand set-up and calibration

**Bionic hand system and set-up.** A custom-made left-hand bionic system was created for this study (Fig. 8). A custom laminated fibreglass socket was fitted around the participant's forearm. The socket includes a lamination ring and a coaxial plug (Össur) that interfaces with an i-LIMB Quantum Hand QWD (Össur). An i-LIMB skin active glove (Össur) was worn on the hand. A custom thermoplastic component housed a Coapt Complete Control Gen2 pattern-recognition system (firmware v.1.27, software v.1.1.9; Coapt; https://coaptengineering.com/technology) and rechargeable lithium polymer batteries (model, 704374; battery rating, 7.4 V, 2000 mAh; capacity, 14.8 Wh; Össur). The Complete Control system is a clinically available EMG pattern-recognition system. The thermoplastic component was attached to the carbon fibre socket using Velcro. The socket was tightly positioned around the participant's forearm using Velcro straps. A custom EMG cable attached to the Coapt EMG controller and connected to electrodes on the upper forearm.

An Össur-certified researcher fitted the bionic hand to each participant's left biological arm. To reduce visual differences between groups, the biological hand was bound in an elastic, sewn fabric (Compressogrip). Participants were then fitted with a custom Boclite foam sheet (to reduce skin irritation). To ensure no interference with the EMG signal, before study participation, all participants shaved the hair on the left forearm. Before electrode placement, the skin of the forearm was cleaned with water and a mildly abrasive paste. Once cleaned, eight EMG electrode pairs (biomedical, disposable pre-gelled silver chloride electrodes; GS-26) were placed around the participant's left forearm. When electrodes were poorly gelled, additional electrode conductive gel was applied. The first electrode pair was placed on the axis of the left extensor digitorum muscle; the remaining electrode pairs were placed around the forearm, roughly equidistant from each other with a preference for sites with optimal muscle contact. In addition, a single reference electrode was placed on the elbow,

above the olecranon bone. To hold the electrodes in place throughout the training, a sweatband was placed around the electrodes. Markings on the skin were used to ensure stable electrode positioning across sessions.

**Calibration protocol.** Bionic hand gestures were introduced to participants serially over the training sessions. On D1, participants calibrated the EMG controller on the bionic open and close gestures. On D2, participants added bionic pinch to their controller. On D3, participants added a bionic tripod to their controller.

At the beginning of training D1, participants practiced making natural muscle contractions for the first two hand gestures in each respective control strategy (biomimetic, open and close; arbitrary, one finger and two fingers). During calibration, participants were instructed to execute hand gestures at 20% of their maximum voluntary contraction (for example, 'would you be able to continue to use this level of force 1,000 times in the session?'). Using the Coapt Complete Control system (Gen2), participants calibrated their EMG controller by serially performing each gesture, guided by the experimenter. This auto-calibration process recorded EMG data during the muscle contractions for each gesture and auto-segmented and auto-labelled the EMG data for each bionic hand gesture class. To maximize the generalizability of the calibration data to the training tasks, we used Coapt's Adaptive Advance feature[51,52], which implements a layering-like algorithm to combine multiple sets (layers) of training data for each gesture. Therefore, we added 7 layers of training data for each gesture: 3 layers with the arm positioned in front of the participant at a 90° angle, 1 layer with the arm positioned to the left, 1 layer with arm positioned to the right, 1 layer with the arm positioned upright and 1 layer with the arm positioned back at baseline. The Coapt system used established classification parameters including 200 ms analysis windows with a 25 ms update increment, time domain and auto-regressive features extracted from each window. Subsequent data were then classified by a linear discriminant analysis classifier[53,54]. Bionic hand velocity control was proportional to the EMG activity[55]. However, the intention to move is constrained by the motors of the bionic hand, creating a delay.

**Training protocol**

Between D1 and D4, participants completed a series of tasks designed to quantify different aspects of bionic hand skill: speed, dexterity and gesture switching. For all tasks, participants' training performance was filmed for an experimenter to perform an offline analysis of all relevant measures.

**Speed (SHAP).** At the beginning of every training session, participants completed a modified version of the SHAP[56]. Participants were instructed to transfer an object as quickly as possible from one position to another. After each transfer, the experimenter would return the object to the starting position. On D1, participants performed 20 speed transfers of lightweight and heavyweight cylinder objects over 6 cm using the open and close bionic hand gestures. On D2, participants repeated the same trials as in D1, and 20 transfers of lightweight and heavyweight 'tip' objects over 5 cm using the open and pinch bionic hand gestures. On D3, participants repeated the same trials as D2, and then 20 transfers of the lightweight and heavyweight 'tripod' objects over 5 cm using the bionic hand open and tripod gestures. On D4, participants repeated the same trials as in D3, and then 20 transfers of the lightweight tripod objects using the bionic hand open and tripod gestures. On the generalization session, participants performed 20 speed transfers of the lightweight cylinder object using the bionic hand open and close gestures

One participant's speed data were excluded due to a technical issue with their Coapt EMG controller. This resulted in their control speed being over 3 s.d. slower than the group average.

**Dexterity (virtual eggs test).** The virtual eggs test is a modified version of the box and blocks test[57]. It involves fragile blocks (that is, eggs[18,58]). The virtual eggs (40 mm × 40 mm × 40 mm, ~80 g) exploit a magnetic fuse mechanism that collapses (that is, breaks) when grasped with a grip force larger than a specific threshold. The break point was calibrated at a force value that was roughly 6 N.

During the task, participants were instructed to transfer eggs over a 20-cm-tall wooden wall as fast as possible without breaking them. The task was only performed using the open and close bionic hand gestures. Participants were encouraged to prioritize grasping the eggs successfully over speed. Participants were told that if the egg broke on initial grasp, they were required to still complete the transfer. Performance was measured as both the number of unbroken (successful) eggs and the number of total attempted eggs transferred within a 2 min time period. Before starting the task, the bionic hand position was pre-set into the open power grip with the thumb manually rotated to be parallel with the other digits. Participants were allowed to practice transferring one egg.

To quantify pressure applied by the bionic hand, we fitted a custom glove on the bionic hand, which included FlexiForce sensors (B201-M-8; Tekscan; https://www.tekscan.com/products-solutions/force-sensors/b201) on the thumb and index finger pads. The sensor had a 0.2 mm thickness and a sensing diameter of 9.7 mm. The force sensor was calibrated, based on manufacturer recommendations, by acquiring a linear calibration curve over a range of forces (10 g, 50 g, 100 g, 200 g, 300 g, 500 g and 1,000 g). The sensor has a 0.375 in. sensing area diameter. We used FlexiForce ELF handles (Tekscan) to transfer the pressure data from the sensors to a laptop. Pressure data were recorded from each of these sensors using the ELF System (Tekscan). At the beginning of the virtual egg test, we applied a trigger signal to the pressure sensors to use as a starting point in the pressure data. For the analysis, the pressure data were averaged across the thumb and index pressure data. Applied pressure was computed as the average pressure during the 2 min task period.

**Gesture switching (block stacking).** Participants had to learn how to engage the gesture-switching functionality of the bionic hand. To switch into a new gesture, participants would first need to engage an open hand signal. This would automatically trigger the bionic hand to move into a baseline hand-open position, ready to switch. Participants could then perform a muscle contraction associated with the desired bionic gesture (close, pinch or tripod) they wanted to switch into. If a short sequence of the correct signal was sent, exceeding a gesture selection confidence threshold, the bionic hand would automatically switch into the open version of the desired bionic gesture and lock into that gesture until switched again. Any maintained signal of the grasping gesture would close the bionic hand proportionally into that closed version of that locked gesture.

To quantify the ability for users to successfully perform gesture switching, we designed a block-stacking task that required participants to grab blocks using pre-defined bionic hand gestures. There were two variations of the task: the two-gesture version required participants to switch between close and pinch, and the three-gesture version required participants to switch between close, pinch and tripod. Participants performed the 2-gesture version on D2, D3, and D4, and the 3-gesture version only on D3. Before starting the task, participants were instructed to grab, transfer and stack blocks (large blocks, 2 in. × 2 in. × 2 in.; small blocks, 1 in. × 1 in. × 1 in.) into towers of 3, as quickly as possible. There were 4 blocks for each of the gestures being tested (that is, 8 total blocks for the 2-gesture version and 12 total blocks for the 3-gesture version). The blocks were arranged such that participants would have to grab a block with the first gesture (close) and the next block with the next gesture (pinch), and so on. If a participant was in an incorrect gesture, participants were instructed to try again until correct. The task finished when participants had successfully transferred all blocks.

## Pre–post testing protocol

We also used a set of pre–post comparison testing measures assessed before and after training: control automaticity, motor control, classification accuracy and sense of embodiment.

**Automaticity of bionic hand control.** *Cognitive load task*. To assess the cognitive load imposed by bionic hand use, a concurrent numerical cognition task was performed during the first (D1) and last (D4) training sessions. The task was adapted from previous studies[35]. Participants were asked to perform two variations of a block-stacking task. The single-condition task required participants to quickly grab, transfer and stack as many blocks (2 in. × 2 in. × 2 in.) as possible into towers of 3 using the bionic hand. The dual-condition task required participants to perform the same block-stacking task while simultaneously verbally performing a counting task. The counting task required participants to follow along to a set of low-, medium- and high-pitch auditory tones played from a laptop. The tones were presented every 2–4 s in a randomized order, for a total duration of 1 min. Participants started the task with the initial count of '50'. Participants were then instructed to (1) add three to the current number if they heard a high-pitch tone, (2) hold the current count if they heard a medium-pitch tone, or (3) subtract three to the current count if they heard a low-pitch tone. For each sound, participants were instructed to verbally respond. To ensure participants were equally motivated for their motor and counting performance, participants were told that their performance would be scored equally on the number of blocks transferred and their counting performance. The primary measures we analysed were the number of blocks participants transferred in the single- and dual-condition tasks (separately) and their counting accuracy (that is, how many trial counts were correct).

To obtain a baseline for counting performance irrespective of motor performance, participants first performed the counting task without any block stacking. Next, participants performed the dual-condition task. Finally, to obtain a baseline for motor performance without cognitive load, participants performed the single-condition (block stacking only) task. In this latter condition, the counting sounds were still played throughout the task; however, participants were told to ignore them.

For each participant, we first calculated the total number of blocks transferred in the single- and dual-condition tasks (separately). To quantify how cognitively demanding the numerical cognition task was on bionic hand motor performance, a control automaticity ratio was computed by dividing the number of blocks transferred in the dual-condition task by the number of blocks transferred in the single-condition task. Counting performance was computed by taking the percentage of total correct mathematical operations.

*Control difficulty questionnaire*. At the end of every session, participants were asked to respond to the following question: 'how difficult was it to control the prosthesis? Please rate between 0 (I found it as easy to perform the movement as using my own hand) to 10 (the most difficult thing imaginable).'

**Motor control.** To quantify participants' motor control for both their left biological hand and the bionic hand, participants performed a ballistic reaching task during the familiarization session and the last (D4) training session (Supplementary Fig. 1). Untrained participants completed this task during the familiarization session and the beginning of the generalization session.

Participants were seated at a custom-made wooden tabletop placed above a digitizing tablet (42.6 cm × 28.4 cm, Intuos Pro Large; Wacom) and facing an LCD monitor (15.6 in., 1,920 pixel × 1,080 pixel dimension; Dell Precision 3560). The participants performed reaching movements by sliding a digitizing stylus (Wacom Pro Pen 3D; Wacom) across the tablet. The position of the stylus was recorded by the tablet at 60 Hz. The experimental software was custom written in python for PsychoPy (v.2021.1.1). Direct vision of the arms (elbows and shoulders included) was occluded using a black barber cape. In addition, the lights were extinguished in the room to minimize peripheral vision of the hand.

Participants performed centre-out planar reaching movements to visual targets. Due to the time constraints with fitting and removing the bionic hand system, all participants performed the task first with their biological hand and then with the bionic hand. Before starting the task, the experimenter locked the bionic hand around the digitizing stylus so that it was immoveable (could not open) for the task. Participants were shown their hand position, the home location and, during trials, the reach target. The hand position was constantly shown to participants (60 Hz), indicated by a green crosshair (0.36 cm × 0.36 cm). The home location was constantly shown to participants as a square (0.36 cm × 0.36 cm) at the bottom, centre of the screen (1.8 cm above the bottom of the screen). The home location was coloured grey between trials and red during trials. The reach target was a white circle (0.18 cm radius) that would appear at 3 separate locations (left, centre or right). The left and right targets were 67° from the vertical centre. All targets were 12.8 cm away from the home location. We also displayed the trial number in the top right corner of the monitor.

Participants completed ten practice trials before starting the task. Participants completed 60 experimental trials. A trial was initiated once participants hovered the cursor over the home location. The home location would then turn red to denote that a reach target would soon appear in one of three target locations. After 3 s, a reach target would appear in one of the 3 locations. Participants were instructed to perform a fast, ballistic reaching movement (within 2 s) towards the centre of each target and to avoid corrective movements, such that they should maintain the end position of their reach until the target disappeared. In total, each trial lasted 5 s. Participants were then required to return to the home location to begin the next trial. The 3 reach targets were each presented 20 times. The order of the targets was pseudorandomized, such that each target was randomly sampled in batches of three. All participants were presented with the same trial order.

Due to technical issues, we excluded the first experimental trial for all participants (that is, 59 total experimental trials). In addition, trials were excluded where the reach end point was above or below 2 s.d. of the participant's reach error (range of excluded trials across participants was 1–6% of total trials).

**Classification accuracy.** To quantify participants' classification accuracy, we used the real-time signal classification output from the Coapt EMG controller (Gen2). Participants were seated at a table, facing an LCD monitor (15.6 in., 1,920 pixel × 1,080 pixel dimension; Dell Precision 3560). Active control of the bionic hand was turned off. Although the bionic hand could not move, the EMG controller was still turned on. On the monitor, participants were shown their real-time signal classification (frame rate, 50 ms), listed as the words 'open', 'close', 'pinch' or 'tripod', or no output, indicating 'rest'.

Participants were instructed to perform and maintain 6 different hand gestures each for 20 s. On the monitor, participants were given a visual indication of the start and end of each gesture trial, and a virtual expanding circle proportional to their level of contraction (In the Zone, a Coapt virtual game). Participants were instructed to perform and maintain each gesture such that they could maintain correct classification, on the monitor, for each of the desired gestures for the duration of the trial. When assessing classification accuracy for three-gesture classes, the trial order was rest, open, close, rest, open and close. When assessing classification accuracy for five-gesture classes, the trial order was rest, open, close, rest, pinch and tripod. Note that when running this task, open bionic gesture trials would terminate after 2.5 s if correct classification and a specified force level was maintained (due to unrelated purposes for a separate study).

The task output file included (1) the real-time classification decision, (2) the trial number the participants were on and (3) the EMG activity for each of the 8 channels (all updated every 50 ms). Previous research has taken a variety of approaches to define a relevant time window for offline analysis of classification accuracy[59]. As open trials would sometimes terminate early (as noted in the previous paragraph), we used an analysis approach that could control for differences in overall trial duration between gestures. We opted to use the first correct classification point, for each gesture trial, as time point zero. Therefore, the average classification accuracy was computed from time point 0 to 49 (approximately the first 2.5 s of each 20 s gesture). This analysis approach was performed for each gesture separately for each participant. On the three-gesture version of the task, where participants were asked to perform each gesture once (that is, two trials per gesture), classification accuracy was calculated separately for each trial of the same gesture and then those values were averaged. To construct the group-level confusion matrices in Fig. 5, values were then averaged across participants for each group. To compute a single value of average classification (Fig. 2d,f), we took the average correct classification across gestures (that is, diagonal of confusion matrix).

Due to technical issues at the beginning of the study, we were not able to acquire these data from the first four study participants (three in the arbitrary and one in the biomimetic group).

**EMG gestural structure.** To quantify the similarity of the EMG patterns for the gestures used in each control strategy, we analysed the EMG data participants generated when training their classifier. There are multiple EMG features used to make a classification decision, as described in 'Calibration protocol'. One of these features is the EMG mean relative value (MRV). To generate a visualization of the EMG profiles, we used the average EMG MRV, computed for each gesture and each channel, during gesture calibration. These data were outputted when each user completed the study. We visualized the MRV for each channel, each gesture and each participant (Fig. 6a). We also averaged the MRV values across participants of each group to visualize the group-level EMG profile. To quantify the similarity observed in this visualization, we computed the euclidean distance in the EMG MRV values across channels for all gesture pairs.

**Sense of embodiment questionnaire.** To assess changes in the sense of embodiment over the bionic hand, participants were asked to complete an embodiment questionnaire before the first and after the last training session (Supplementary Table 1). Untrained participants filled out the post-questionnaire after completing the post-motor control assessments, before being able to experience active control of the bionic hand. The questionnaire was focused on the explicit (phenomenological) aspect of embodiment, whether the bionic hand feels like a part of one's body. Participants were asked to rate their agreement with 10 statements[35,60] on a 7-point Likert-type scale ranging from −3 (strongly disagree) to +3 (strongly agree). Statements were clustered into three main categories, probing different aspects of embodiment: body ownership, agency and body image. For each participant, questionnaire scores were averaged within each embodiment category. To compute a difference score, pre-scores for each embodiment category were subtracted from the post-scores.

**Statistical analyses**

All statistical analyses were performed using JASP (v.0.14). Tests for normality were conducted using a Shapiro–Wilk test. When assumptions of normality were met, we used parametric statistics, and when they were not met ($P < 0.05$ for the Shapiro–Wilks test), equivalent non-parametric tests were used. Between-group comparisons were conducted using repeated-measures analysis of variance with group (biomimetic, arbitrary or untrained) as a fixed effect and two-tailed independent samples $t$-tests (parametric) or two-tailed Mann–Whitney

tests (non-parametric). Within-group comparisons were conducted using two-tailed paired $t$-tests (parametric) or two-tailed Wilcoxon tests (non-parametric). All non-significant results were further examined using corresponding Bayesian tests, represented as a Bayes factor ($BF_{10}$), under continuous prior distribution (Cauchy prior width $r = 0.707$). We interpreted the test based on the well-accepted criterion of Bayes factor smaller than one-third as supporting the null hypothesis[61].

To investigate whether cognitive (control automaticity) or EMG (euclidean distance between first two trained gestures) factors drove the group differences we observed on the SHAP, we added each measure as a regressor to the SHAP speed scores. The regression model was estimated using ordinary least squares. We then tested for group differences in the unstandardized residuals of the SHAP D1 scores when controlling for each measure.

**Reporting summary**

Further information on research design is available in the Nature Portfolio Reporting Summary linked to this article.

## Data availability

Pre-registered study predictions and methods, and the data used in the study, can be accessed at https://osf.io/3m592/. It is important to note that we deviated slightly on some planned analyses in the pre-registration; for example, for an independent measure of motor control, we opted to use the target-reaching data and not the circle-tracing data, as performance for all participants was much noisier than anticipated due to the physical apparatus of the task design. Lastly, the pre-registration also included a description and planned analyses for the functional magnetic resonance imaging portion of the study, which will be reported in a separate manuscript.

## Code availability

Code used in the study can be accessed at https://github.com/hunterschone/ProControl.

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

## Acknowledgements

C.I.B. and H.R.S. are supported by the Intramural Research Program of the National Institute of Mental Health (ZIAMH 002893). T.R.M. was supported by the European Research Council (715022 Embodied Tech). The funders had no role in study design, data collection and analysis, decision to publish or preparation of the article. We thank J. Ingelholm for technical support; M. Controzzi and F. Clemente for sharing the original design of the virtual eggs; L. Emmrich (Össur) for consultation on the bionic hand system design and training paradigm; A. Moore (Össur) and N. Wagner (Össur) for fabrication of the bionic hand system; L. Teichmann for providing technical and conceptual support throughout the study; the OP4 clinic nursing staff for their patience and support for our 60 participants; M. Reel, our nurse practitioner, for his commitment and patience with all the participant visits; the Medical Center Orthotics and Prosthetics (MCOP) for help in device fabrication and technical support; and the technical support team at Coapt for providing regular technical support for the EMG controllers. T.R. Makin was supported by the European Research Council (715022 Embodied Tech), Wellcome Trust (215575/Z/19/Z) and Medical Research Council (MC_UU_00030/10).

## Author contributions

H.R.S. designed the research, collected the data, analysed all datasets and wrote the first version of the article. T.R.M. and C.I.B. designed the research, supervised analyses and edited the article. M.U. assisted in data collection and analysed the motor control data. M.M. assisted in data collection and analysed the classification accuracy data. B.R. assisted in data collection. J.V., B.L. and L.H. provided technical support and edited the manuscript.

## Competing interests

B.L. and L.H. have a financial interest in Coapt LLC (https://www.coaptengineering.com), which manufactures a device component being used in this research. The remaining authors declare no competing interests.

## Additional information

**Correspondence and requests for materials** should be addressed to Hunter R. Schone or Tamar R. Makin.

# Reporting Summary

## Statistics

For all statistical analyses, confirm that the following items are present in the figure legend, table legend, main text, or Methods section.

| n/a | Confirmed | |
|---|---|---|
| ☐ | ☒ | The exact sample size (*n*) for each experimental group/condition, given as a discrete number and unit of measurement |
| ☐ | ☒ | A statement on whether measurements were taken from distinct samples or whether the same sample was measured repeatedly |
| ☐ | ☒ | The statistical test(s) used AND whether they are one- or two-sided *Only common tests should be described solely by name; describe more complex techniques in the Methods section.* |
| ☐ | ☒ | A description of all covariates tested |
| ☐ | ☒ | A description of any assumptions or corrections, such as tests of normality and adjustment for multiple comparisons |
| ☐ | ☒ | A full description of the statistical parameters including central tendency (e.g. means) or other basic estimates (e.g. regression coefficient) AND variation (e.g. standard deviation) or associated estimates of uncertainty (e.g. confidence intervals) |
| ☐ | ☒ | For null hypothesis testing, the test statistic (e.g. *F*, *t*, *r*) with confidence intervals, effect sizes, degrees of freedom and *P* value noted *Give P values as exact values whenever suitable.* |
| ☐ | ☒ | For Bayesian analysis, information on the choice of priors and Markov chain Monte Carlo settings |
| ☐ | ☒ | For hierarchical and complex designs, identification of the appropriate level for tests and full reporting of outcomes |
| ☐ | ☒ | Estimates of effect sizes (e.g. Cohen's *d*, Pearson's *r*), indicating how they were calculated |

*Our web collection on statistics for biologists contains articles on many of the points above.*

## Software and code

Policy information about availability of computer code

| Data collection | EMG data was recorded using the Coapt COMPLETE CONTROL Gen2 pattern recognition system [Coapt, LLC; firmware v1.27; software v.l.1.9]. Presentation software included PsychoPy (v2021.1.1). |
|---|---|
| Data analysis | All statistical analyses were performed using JASP (v0.14). All data was analyzed using custom Python (version 3) scripts. Code used in the study can be accessed at https://github.com/hunterschone/ProControl. All code will be made available prior to publication. |

For manuscripts utilizing custom algorithms or software that are central to the research but not yet described in published literature, software must be made available to editors and reviewers. We strongly encourage code deposition in a community repository (e.g. GitHub). See the Nature Portfolio guidelines for submitting code & software for further information.

## Data

Policy information about availability of data

All manuscripts must include a data availability statement. This statement should provide the following information, where applicable:
- Accession codes, unique identifiers, or web links for publicly available datasets
- A description of any restrictions on data availability
- For clinical datasets or third party data, please ensure that the statement adheres to our policy

Pre-registered study predictions and methods, as well as the data used in the study, can be accessed at: https://osf.io/ 3m592/.

# Research involving human participants, their data, or biological material

Policy information about studies with <u>human participants or human data</u>. See also policy information about <u>sex, gender (identity/presentation), and sexual orientation</u> and <u>race, ethnicity and racism</u>.

| | |
|---|---|
| Reporting on sex and gender | Sixty-one healthy volunteers (40 females; mean age= 24.8 ± 0.66; all right handed) were recruited from the National Institute of Health community and the Washington DC metro area and were randomly assigned to one of the following study groups: biomimetic (n = 21; 14 females; mean age 23.9 ± 0.57), arbitrary (n = 21; 12 females; mean age 25.9 ± 1.28) or untrained (n = 19; 14 females; mean age 24.6 ± 1.41). Information on sex was self-reported by the volunteers. |
| Reporting on race, ethnicity, or other socially relevant groupings | *Please specify the socially constructed or socially relevant categorization variable(s) used in your manuscript and explain why they were used. Please note that such variables should not be used as proxies for other socially constructed/relevant variables (for example, race or ethnicity should not be used as a proxy for socioeconomic status).*<br>*Provide clear definitions of the relevant terms used, how they were provided (by the participants/respondents, the researchers, or third parties), and the method(s) used to classify people into the different categories (e.g. self-report, census or administrative data, social media data, etc.)*<br>*Please provide details about how you controlled for confounding variables in your analyses.* |
| Population characteristics | Sixty-one healthy volunteers (40 females; mean age= 24.8 ± 0.66; all right handed) were recruited from the National Institute of Health community and the Washington DC metro area and were randomly assigned to one of the following study groups: biomimetic (n = 21; 14 females; mean age 23.9 ± 0.57), arbitrary (n = 21; 12 females; mean age 25.9 ± 1.28) or untrained (n = 19; 14 females; mean age 24.6 ± 1.41). Information on sex was self-reported by the volunteers. All participants had no known motor disorders, as determined by a nurse practitioner. |
| Recruitment | All participants were recruited from the National Institute of Health community and the Washington DC metro area via an NIH research participant online database and word of mouth. |
| Ethics oversight | The study and its experimental procedures were approved by the NIH Institutional Review Board (NCT00001360, 93M-0170). |

Note that full information on the approval of the study protocol must also be provided in the manuscript.

# Field-specific reporting

Please select the one below that is the best fit for your research. If you are not sure, read the appropriate sections before making your selection.

☐ Life sciences  ☒ Behavioural & social sciences  ☐ Ecological, evolutionary & environmental sciences

For a reference copy of the document with all sections, see nature.com/documents/nr-reporting-summary-flat.pdf

# Behavioural & social sciences study design

All studies must disclose on these points even when the disclosure is negative.

| | |
|---|---|
| Study description | Quantitative experimental |
| Research sample | Sixty-one healthy volunteers (40 females; mean age= 24.8 ± 0.66; all right handed) were recruited from the National Institute of Health community and the Washington DC metro area and were randomly assigned to one of the following study groups: biomimetic (n = 21; 14 females; mean age 23.9 ± 0.57), arbitrary (n = 21; 12 females; mean age 25.9 ± 1.28) or untrained (n = 19; 14 females; mean age 24.6 ± 1.41). The sample size is based on a power analysis for learning effects with an artificial body-part (Kieliba et al., 2021, Sci Robotic) and was specified in the studies pre-registration. |
| Sampling strategy | Our sample sizes were based on a power analysis for learning effects with an artificial body part, (see Kieliba et al., 2021, Sci. Robotics), as well as what was technically feasible to accommodate the most number of sessions for the most number of subjects. |
| Data collection | There are a variety of data-types included in the study. EMG data was recorded using the Coapt COMPLETE CONTROL Gen2 pattern recognition system (Coapt, LLC; firmware vl.27; software v.l.1.9). Motor control data was collected using a digitizing tablet (42.6 by 28.4 cm, lntuos Pro Large; Wacom, Vancouver, WA). All bionic hand training sessions were filmed. Bionic hand task data (speed, dexterity, gesture switching, control automaticity) was recorded by pen and paper and validated offline by video analyses. For all sessions, a single researcher and the research participant were present. Because the testing required the researcher to know which control strategy each participant belonged to, the researcher could not be blind to which training group a participant belonged to. |
| Timing | All data collection took place between May 3, 2021 to May 2nd, 2022. |
| Data exclusions | One participant's speed data was excluded, due to a technical issue with their Coapt EMG controller. |
| Non-participation | Two additional participants were recruited, but not included in the present study due to dropping out prior to competing the study. |
| Randomization | Volunteers were randomly assigned to the study groups. |

# Reporting for specific materials, systems and methods

We require information from authors about some types of materials, experimental systems and methods used in many studies. Here, indicate whether each material, system or method listed is relevant to your study. If you are not sure if a list item applies to your research, read the appropriate section before selecting a response.

## Materials & experimental systems

| n/a | Involved in the study |
|-----|----------------------|
| ☒ ☐ | Antibodies |
| ☒ ☐ | Eukaryotic cell lines |
| ☒ ☐ | Palaeontology and archaeology |
| ☒ ☐ | Animals and other organisms |
| ☒ ☐ | Clinical data |
| ☒ ☐ | Dual use research of concern |
| ☒ ☐ | Plants |

## Methods

| n/a | Involved in the study |
|-----|----------------------|
| ☒ ☐ | ChIP-seq |
| ☒ ☐ | Flow cytometry |
| ☒ ☐ | MRI-based neuroimaging |

## Plants

Seed stocks — *Report on the source of all seed stocks or other plant material used. If applicable, state the seed stock centre and catalogue number. If plant specimens were collected from the field, describe the collection location, date and sampling procedures.*

Novel plant genotypes — *Describe the methods by which all novel plant genotypes were produced. This includes those generated by transgenic approaches, gene editing, chemical/radiation-based mutagenesis and hybridization. For transgenic lines, describe the transformation method, the number of independent lines analyzed and the generation upon which experiments were performed. For gene-edited lines, describe the editor used, the endogenous sequence targeted for editing, the targeting guide RNA sequence (if applicable) and how the editor was applied.*

Authentication — *Describe any authentication procedures for each seed stock used or novel genotype generated. Describe any experiments used to assess the effect of a mutation and, where applicable, how potential secondary effects (e.g. second site T-DNA insertions, mosaicism, off-target gene editing) were examined.*

