## [Peer Review File · Nature Human Behaviour]

Peer Review Information

Journal: Nature Human Behaviour

Manuscript Title: Biomimetic versus arbitrary motor control strategies for bionic hand skill learning

Corresponding author name(s): Hunter R. Schone

Reviewer Comments & Decisions:

Decision Letter, initial version:

3rd April 2023

Dear Mr Schone,

Thank you once again for your manuscript, entitled "Should bionic limb control mimic the human body? Impact of control strategy on bionic hand skill learning," and for your patience during the peer review process.

Your manuscript has now been evaluated by 4 reviewers, whose comments are included at the end of this letter. Although the reviewers find your work to be of interest, they also raise some important concerns. We are interested in the possibility of publishing your study in Nature Human Behaviour, but would like to consider your response to these concerns in the form of a revised manuscript before we make a decision on publication.

To guide the scope of the revisions, the editors discuss the referee reports in detail within the team, including with the chief editor, with a view to (1) identifying key priorities that should be addressed in revision and (2) overruling referee requests that are deemed beyond the scope of the current study. We hope that you will find the prioritized set of referee points to be useful when revising your study. Please do not hesitate to get in touch if you would like to discuss these issues further.

Although the reviewers appreciate several aspects of your work, they are generally in agreement that the current work does not answer the question it poses. Editorially, this raises concerns regarding the suitability of this work for publication in Nature Human Behaviour. Therefore, in a revised version it would be essential for you to provide additional data - either behavioral or neural - to elucidate the mechanisms and provide a more definitive response to the main research question. We appreciate that this request for additional data is a tall order, and you may prefer to pursue the publication of this work with its existing data and more circumscribed conclusions elsewhere. In that case, we would be happy to facilitate potential transfer to other journals in the Nature Portfolio or elsewhere.

If you wish to submit a suitably revised manuscript, we would hope to receive it within 4 months. I would be grateful if you could contact us as soon as possible if you foresee difficulties with meeting this target resubmission date.

- Include a "Response to the editors and reviewers" document detailing, point-by-point, how you addressed each editor and referee comment. If no action was taken to address a point, you must provide a compelling argument. When formatting this document, please respond to each reviewer comment individually, including the full text of the reviewer comment verbatim followed by your response to the individual point. This response will be used by the editors to evaluate your revision and sent back to the reviewers along with the revised manuscript.
- Highlight all changes made to your manuscript or provide us with a version that tracks changes.

[REDACTED]

Thank you for the opportunity to review your work. Please do not hesitate to contact me if you have any questions or would like to discuss the required revisions further.

Sincerely,

Giacomo Ariani
Editor
Nature Human Behaviour

Reviewer expertise:

Reviewer #1: Sensorimotor Rehabilitation, Embodiment

Reviewer #2: Neuroprosthetics, Sensorimotor Rehabilitation, BMI

Reviewer #3: Robotics, Sensorimotor Rehabilitation, BMI

Reviewer #4: Biomechanics, Prosthetics, Sensorimotor Rehabilitation

REVIEWER COMMENTS:

Reviewer #1:

Remarks to the Author:

Schone and colleagues compared learning and performance of different motor tasks executed by healthy participants while controlling a wearable myoelectric prosthetic hand, either via a biomimetic or an arbitrary control strategy. The key results are that while learning is faster (for some parameters) and less cognitively demanding in the initial phase with the biomimetic control, generalization to novel tasks is better for the arbitrary control strategy.

The results are discussed in terms of strategy to develop future prostheses control in amputees.

The paper is excellently written and the results are clear.

My main question is about the EMG data. The authors analyzed at the beginning of the results section the classifier performance, to show no difference between groups. But I guess the different performance between the biomimetic and the arbitrary control in the first sessions should be seen in some EMG parameters. Could the authors dive more deeply into these data to correlate changes in EMG-based control with performance? This would contribute to providing some understanding of the underlying effects.

Then I have a few remarks on the way results are interpreted to convey the authors' message.

Page 4, Embodiment question. The authors wrote:

"Overall, contrary to the common assumptions of biomimetic design, biomimetic control didn't provide an increased sense of embodiment."

The authors should mitigate this statement, as it could be that the study doesn't have enough power to show a significant effect. Actually, in terms of ownership, there is a trend towards higher embodiment. Add a "statistically significant" at least in the sentence.

Then, more in general, the main claim of the paper is that these results should be generalized to the field of prosthetics. I am not sure that this step is so straightforward. There is a big difference between adding and controlling an additional body part and replacing a missing one. It could be that biomimetic control has much more value for replacing than adding an artificial limb, as tested in a rather specific and ad-hoc context as the one realized here. This needs to be acknowledged, or discussed if the authors do not agree.

Page 10:"For example, learning with a more varied input that increases task difficulty is initially slower, but typically yields better generalization [for a review see (37)]. Accordingly, research has demonstrated that more complex associations end up being learned more robustly and are better retained than easy-to-learn associations (38)."

This is a key point to explain the mechanism underlying the effects reported here. It needs to be more precisely discussed and supported either with more straightforward concurrent evidence or better with further analyses of the current data, or at least by proposing further experiments to test this hypothesis.

Andrea Serino

Reviewer #2:

Remarks to the Author:

Schone and colleagues sought to determine whether biomimetic motor control enhances device embodiment, learning, generalization, and automaticity in anthropomorphic bionic limbs. This is very important to enhance prosthetic functionality and user experience of current and future arm prostheses.

The researchers compared biomimetic and non-biomimetic control strategies, with a few well-thought additional controls, by testing able-bodied participants operating a wearable myoelectric bionic hand and providing them an array of user experience tests, falling in two categories: skill tasks and questionnaires.

I applaud the authors for the extremely well-crafted experimental design, which is indeed representative of today's commercial technology. The article is very freshly written and engaging. The pop references are on point! The figures are perfectly clear, although I would suggest that the authors (as they have done in panel A of Figure 2) would replicate the approach of schematizing the presented test in Fig. 4 and 5.

There is a weakness of the article that I wish to discuss, which is also at least in part expressed by the authors as a limitation in the discussion. The experimental design applies to myoelectric prostheses which are slow and actuated by EMG pattern recognition. This is a control bottleneck, that forces the user to 1) substantially anticipate planning knowing they will receive a lagged feedback (= a substantial dose of feed-forward control) and 2) reduce the dimensionality of degrees of freedom in 'natural' control to pass through the bottle-neck of a few available patterns.

Now, this is exactly how current myoelectric prostheses work. They are mostly slow and bound to a few patterns. This research applies to that egregiously.

Yet, the question at the forefront is "Should bionic limb control mimic the human body?", we are taking a generalization step back to look at defining general laws of human control of bionic limbs. The manuscript is indeed constructed around this very important discussion, but can only address a portion of it, due not to flaws in research design, but to flaws of current technology.

I suggest that a more precise definition of the tested hypothesis would be obtained by putting upfront the application to myoelectric pattern recognition prostheses, as a limited proxy of the more general framework.

As of the higher question, a thought experiment I suggest is the following: what if we were to redo

these experiments but using video recognition of hand movements and having to control a virtual hand avatar. Suppose that the control lag is very small, e.g., <50ms. Would you expect not to see a persistent lag of arbitrary strategies with respect to purely biomimetic? Even in light of the current results, I still would expect a performance lag. Perhaps I am wrong, but the question is not resolved and open to discussion and hypotheses.

The thought experiment shows that there will be a spectrum of control fidelities to which the general theory applies. Here we are on the lower fidelity side (which are currently state of the art) and the results stand, but better control fidelities may lead to a biomimetic advantage.

In summary, I propose 1) to tone down the overall claim to better fit the application sphere of current pattern recognition myoelectric prostheses and 2) to acknowledge that different control fidelities may lead to a different ending balance when comparing the two approaches.

Reviewer #3:

Remarks to the Author:

The authors present a rigorous study addressing an important topic and open question in the field. The manuscript is suitable for publication and would certainly be of interest to the community. That said, the impact of the manuscript is limited in two ways and some claims in the paper should be softened in recognition of these limitations:

1) All tasks can be broken down into a single and simple fundamental challenge that does not address the full span of motor coordination or hand dexterity. In essence, all tasks simply require the user to perform a muscle pattern to activate a grasp, move their forearm, and then perform another muscle pattern to terminate the grasp. Should biomimetic control strategies provide benefits, it's likely these benefits would be observed when tasks are difficult (e.g., with a novel or a complex task). Indeed, data from the manuscript support this idea – as the initial attempts at a task (i.e., when the task is the most novel and complex) have biomimetic control performing better. Ideally, the authors would have explored this idea more fully. A more appropriate test would be to compare biomimetic vs arbitrary control on increasingly difficult tasks that involve more motor coordination (e.g., using simultaneous and proportional control of multiple degrees of freedom to complete the same tasks, and more difficult, novel tasks, such as the harder tasks the authors omitted from the SHAP test). That said, I recognize the limitations of EMG control to pursue such an ideal comparison. Instead, I suggest the authors address this limitation in the manuscript and consider exploring this in future work. It would also be appropriate to highlight that the data show biomimetic control consistently outperforming arbitrary control when tasks are difficult (i.e., the first time users completed a task). It remains unclear if arbitrary users could scale performance to biomimetic users when tasks become much more difficult; there may only be so much capacity for an individual to learn a new unintuitive control strategy, such that as the complexity of the hand increases, arbitrary control could lag.

2) In a somewhat similar light, generalization of control strategies should be addressed with a novel task, not a novel arbitrary control strategy. It is somewhat trivial that arbitrary users can generalize better to arbitrary control strategies. For the claims the authors want to make regarding control generalization, they should have performed a novel task instead of a novel control strategy. Further, what are the real-world equivalents of generalization to a novel control strategy? Healthy individuals

and prosthesis users do not regularly change the fundamental way they are controlling their hand. More accurately, they encounter new tasks that require new coordinated motor actions that they must learn on the fly. A more appropriate test would be to compare biomimetic and arbitrary control on a novel unseen task. Given this limitation, I suggest the authors soften claims regarding the importance of generalization to different arbitrary control strategy, or explain a real-life use-case and the value of generalization from one arbitrary control strategy to another arbitrary control strategy.

Despite the limitations above, the present manuscript makes contributions towards an important question and sets the stage for future work in this area. Some additional comments are below:

- Use of 1 – 4 gestures is not “as far away from biomimetic” as possible. It’s logical and intuitive in that it’s the order the gestures were introduced.
- Not included is preference among control strategies. Patients report wanting bionic hands that are visually similar to human hands. They may also simply prefer biomimetic despite it not necessarily providing optimal control (similar to how patients use non-function cosmetic prostheses).
- No information is provided on the proportional control needed for the VET. Please provide methodology for how the myoelectric classifier provides proportional control. Are users controlling hand position directly? Hand velocity?
- How do the values reported here compare to other uses of the VET, SHAP and mode-switching tasks? Are the users in this study performing much worse or much better overall? This comparison can help assess if performance was simply due to a task being too easy or too difficult.
- What is the ratio of object weight to break force used in the VET test? How does this compare to prior uses of fragile object tasks? Is this an easy version of a fragile object test or a difficult version?

Reviewer #4:

Remarks to the Author:

This is a well written paper describing an interesting study and experimental paradigm. I have just a few comments. First, the lack of ‘generalizability’ does not necessarily seem surprising as in one case, users are trained on an arbitrary mapping and the other case a biomimetic mapping. In the generalization, they are trained in a different arbitrary mapping – which would seem to bias learning toward the first group. It might be interesting instead to swap control approaches at this phase (i.e., arbitrary then does biomimetic). I don’t think this needs to be done at this stage, but the point could be better addressed in the discussion.

Minor comments:

Biomimetic advantage section, paragraph 3 – ‘no significant differences’ should say ‘in’ not ‘on’ (line numbers would help review)

Subsequent section – suggest rephrasing to ‘more affected by the cognitive task’ as cognitively impaired implies something different

Remove ref 42 as paper is not yet submitted

Author Rebuttal to Initial comments

REVIEWER COMMENTS:

Reviewer #1:

- 1.1 *“Schone and colleagues compared learning and performance of different motor tasks executed by healthy participants while controlling a wearable myoelectric prosthetic hand, either via a biomimetic or an arbitrary control strategy. The key results are that while learning is faster (for some parameters) and less cognitively demanding in the initial phase with the biomimetic control, generalization to novel tasks is better for the arbitrary control strategy. The results are discussed in terms of strategy to develop future prostheses control in amputees. The paper is excellently written and the results are clear.”*

We first want to thank the reviewer for taking the time to read and evaluate the manuscript.

- 1.2 *“My main question is about the EMG data. The authors analyzed at the beginning of the results section the classifier performance, to show no difference between groups. But I guess the different performance between the biomimetic and the arbitrary control in the first sessions should be seen in some EMG parameters. Could the authors dive more deeply into these data to correlate changes in EMG-based control with performance? This would contribute to providing some understanding of the underlying effects.”*

In the original manuscript, we observed that arbitrary users had slower performance than biomimetic users early in their training (**Fig. 2A**), though we did not observe these group differences in EMG classification accuracy (**Fig. 5A-F**), i.e., the EMG controller classifies gestures for both control strategies similarly. So, what could be causing this speed delay for arbitrary users early in their training? The most obvious reason is the clear cognitive disadvantage for arbitrary users early in training (**Fig 4**; e.g., the time to remember and regenerate the relevant movement to actuate the grasp). The other possibility, which the reviewer hints at, is that there could be group differences in the EMG parameters, most notably, the EMG gestural structure. In other words, the arbitrary gestures could be inherently more similar to each other, making it more difficult to precisely execute the necessary muscle contraction, relative to the other gestures, to actuate the desired grasp. The idea we wished to explore, based on the reviewer’s comment, is that even though the EMG classification accuracy is similar across groups, there might still be a greater ‘cost’, be it cognitive or motor, to produce this similar level of performance. Though, these factors: cognitive and motor, are not entirely independent of each other and it is possible that both contribute to the observed group differences.

To explore this, in the revised manuscript, we performed more analyses on the EMG data. In particular, we investigated (1) the gestural similarity in the EMG profiles for each control strategy and (2) the relationship between these EMG measures and Day 1 functional performance.

1) Gestural similarity in the EMG profiles for each control strategy.*Choosing a dataset*

To investigate the gestural similarity in the EMG profiles, there are multiple datasets we collected that we could use: (1) the classification accuracy dataset, which tests a user’s ability

to use their already trained classifier, or (2) the EMG dataset used to train the classifier (for information on this dataset, see the *Methods*, section on *Calibration protocol*). We opted to use the second dataset, because it is the data the controller is exclusively relying on for users to control their device. It's worth noting that we do not have these data for the first few study

participants tested, due to an issue with the initial firmware not saving the EMG classifier data after study completion.

Visualizing the EMG profiles

To investigate the EMG gestural structure for each strategy, we started by visualizing the EMG profile for each trained gesture, for each participant. We first extracted the EMG classifier data for each participant and looked at the EMG amplitudes for each of the 8 EMG channels for each trained gesture. For context, remember that different gestures are trained on different training days: D1 (open and close), D2 (pinch) and D3 (tripod) and the classifier saves these data and compares real-time data to it to make classification decisions. From this visualization, we observed that the group-level EMG profiles showed more similar patterns between the first two trained arbitrary gestures (1 finger and 2 fingers) compared to the first two trained biomimetic gestures (open and close; **Revision Fig. 1A**). This could help explain why the arbitrary group shows slower performance early in training, i.e. they have to work harder to perform and maintain each gesture for control, due to increased similarity between their first two trained gestures.

Quantifying EMG gestural similarity

To quantify this observation, we next computed the euclidean distance between all gesture combinations (**Revision Fig. 1B**). On average, there were no group differences in the average euclidean distance, across channels, between all trained gesture combinations ($W=131.0$, $p=0.346$, $BF_{10}=.73$). To test our hypothesis, whether more similar gestures in the arbitrary strategy impacted Day 1 performance, we exclusively looked at the euclidean distance between the first two-trained gestures (biomimetic: open and close; arbitrary: 1 finger and 2 fingers), i.e., the gestures used during Day 1 performance where we observed group differences. Indeed, we observed that the first two trained gestures for arbitrary users have significantly smaller euclidean distances, i.e. more similar, relative to biomimetic users ($W=250.0$, $p=0.004$). Note that this comparison was specifically chosen to address the behavioral gap we observed, and as such we did not correct the p statistics for multiple comparisons.

EMG data when training the classifier

Revision Figure 1 (Figure 5G-J in revised manuscript). Investigating gestural similarity in the EMG profiles of each control strategy. (A) A visualization of the EMG mean relative value profile for the group average (top) and individual participants (bottom) for each gesture across the 8 surface EMG channels around the forearm (using the EMG data participants generated to train the classifier). Values reflect the average EMG amplitude at each channel when holding the gesture during calibration. **(B)** A matrix filled with the euclidean distances between each gesture pair across channels. **(C)** No differences between trained groups in average euclidean distance across all trained gestures ($W=131.0$, $p=0.346$; $BF_{10}=0.73$). **(D)** A group comparison between EMG profiles for their first two trained gestures between the arbitrary (1 finger and 2 fingers) and biomimetic users (open and close) revealed more distinct patterns in the biomimetic group ($W=250.0$, $p=0.004$). Circles depict individual subject means (across relevant items).

Values indicate group means \pm standard error.

2) Relationship between day 1 EMG measures and day 1 functional performance

Above, we found that there are group differences in the EMG profiles of trained gestures on Day 1. Though, we still don't know whether this is contributing to the group differences in speed on Day 1 or whether it's the cognitive disadvantage or more likely a combination of both factors. To investigate this, we wanted to perform two exploratory spearman correlations comparing SHAP Day 1 speed to (1) the euclidean distance between the first two trained gestures and (2) control automaticity tested on Day 1. These correlations are exploratory and spurious, in that we already know there are group differences on Day 1 in all three measures (Makin and Orban de Xivry, 2019). That said, it's a helpful starting point to characterize the relationship between these measures. From these correlations, we observed somewhat of a relationship between SHAP performance and the euclidean distance between the first two trained gestures (8% of variance explained), though this was non-significant (**Revision Fig. 2**; $r_s=-0.293$, $p=0.08$). In contrast, we observed a significant relationship between control automaticity and SHAP performance, with 23% of the variance explained ($r_s=-0.478$, $p=0.003$).

SHAP control speed on Day 1 versus

Revision Figure 2 (Supp. Figure 2 in revised manuscript). Relationship between Day1

measures. (Left) There was not a significant relationship between SHAP Day 1 control speed and the euclidean distance between the first two trained gestures (used on the task; $r_s=-0.293$, $p=.08$). **(Right)** There was a significant relationship between Day 1 control automaticity and SHAP Day 1 control speed ($r_s=-0.478$, $p=.003$), such that more automatic control reflected faster performance on the SHAP.

To break away from these spurious correlations, we can also control for each measure (euclidean distance or control automaticity) in the SHAP Day 1 speed scores to see whether these factors impact the group differences we observed on the SHAP. To do this, we added each measure as a regressor and tested for group differences in the unstandardized residuals of SHAP Day 1 scores. When controlling for euclidean distance, we still observed a significant group difference in speed ($t(33)=-2.375$, $p=.024$) and when controlling for control automaticity, the difference was not significant ($t(37)=-1.912$, $p=.064$; $BF_{10}=1.281$). The main takeaway from these analyses is that the group differences in speed observed on day 1 are likely due to a combination of the EMG gestural structure of the control strategies and the control automaticity surrounding the strategy mapping, with the latter perhaps playing a greater role.

In conclusion, as discussed in the revised manuscript (see relevant revisions below), our evidence demonstrates that the group differences in performance were restricted to the speed task, and not EMG classification accuracy or proportional control. While we initially attributed this difference strictly to the biomimetic element of the control strategy, a deeper investigation revealed that the euclidean distance between the EMG profile for the first two trained gestures was significantly smaller for the arbitrary control strategy than biomimetic strategy. In other words, arbitrary users would need to execute more precise gestures to achieve the same level of classification accuracy as biomimetic users. It is important to emphasize that this latter consideration is not inherent to the arbitrary approach but is rather a limitation of our own design. Though, considering that EMG gestural structure plays some role in performance, this provides a strong prediction that an arbitrary strategy with gestures with more distinct EMG patterns than biomimetic gestures could theoretically perform better.

In the revised manuscript, we've incorporated these analyses and expand our discussion on these limitations. Additionally, we created a main text figure with this EMG analysis and moved the classification accuracy data that was in previously in Supplementary results into the main text.

In the *Results* section:

Initial biomimetic gestures have more distinct EMG patterns than arbitrary gestures

What could be driving the speed delay for arbitrary users early in their training? The most obvious explanation is the cognitive disadvantage for arbitrary users (e.g. the time it takes to regenerate the relevant movement to actuate the grasp), as seen in the dual cognitive-motor task findings. Alternatively, there could be differences in the EMG gestural structure of the control

strategies (e.g., some gestures could be more difficult to precisely articulate, relative to other gestures). In other words, even though we observed similar EMG classification accuracy across groups (Figure 5A-F), there might still be a greater cost on one group in order to achieve the same level of classification accuracy. To explore this, we investigated the EMG data participants generated when training their classifier. We visualized the EMG profiles across channels at the group and individual subject level (Figure 5G). To quantify the similarity between gestures across EMG channels, we computed the euclidean distance between all gesture combinations (Figure 5H). From this analysis, we observed that across all trained gestures there were no significant group differences (Figure 5I; $W=131.0$, $p=0.346$; $BF_{10}=0.73$). However, for the first two trained gestures (open and close) used on Day 1, arbitrary users have significantly smaller euclidean distances, i.e. more similar, relative to biomimetic users (Figure 5J; $W=250.0$, $p=0.004$). In other words, arbitrary users would need to generate more precise muscle contractions in order to achieve the same level of classification accuracy as biomimetic users (at least for the first two trained gestures).

To further explore whether control automaticity or euclidean distance between gestures drive the group differences we observed on the speed task, we added each measure as a regressor and tested for group differences in the unstandardized residuals of SHAP Day 1 scores. When controlling for euclidean distance, we still observed a significant group difference in speed ($t(33)=-2.375$, $p=0.024$) and when controlling for control automaticity, we observed a trending group difference ($t(37)=-1.912$, $p=0.064$; $BF_{10}=1.281$). These analyses indicate that the group differences in speed observed on Day 1 may be exacerbated by a combination of the EMG gestural similarity of the control strategies and the control automaticity surrounding the strategy mapping, with the latter perhaps playing a greater role (see Supp. Figure 2 for correlations between measures).

EMG data when testing the trained classifier

EMG data when training the classifier

Figure 5. EMG training and testing data. (A) An example visualization of the EMG-signals acquired from the bionic hand system. (B) An example visualization of the real-time EMG signal and classification decision while participants engaged and held each hand gesture for 20 seconds (see Methods). During this assessment, the Coapt controller outputted a real-time

classification decision (updated every 50ms) for which gesture was being performed. Comparing this output to the gesture participants were instructed to perform by the experimenter, we computed the classification accuracy for each gesture. The time-window of interest was approximately the first 2.5 seconds of each 20 second gesture trial (shown in panel E as a red window). (C) Group average classification accuracy matrices for 3-gesture classes (rest, open close) groups assessed on D1, immediately following controller calibration. (D) Both trained groups showed a significant increase in average classification accuracy (main effect of day: $F_{(1,31)}=5.647$, $p=0.024$) between the first (D1) and last day of training (D4). No significant differences were found between the two training strategies before training ($W=167.0$, $p=0.631$; $BF_{10}=0.37$) or after training ($W=121.50$, $p=0.437$; $BF_{10}=0.37$). (E) Group average classification accuracy matrices for 5-gesture classes (rest, open close, pinch, tripod), assessed on D4, at the end of training. (F) No differences between trained groups in average classification accuracy (average of the 5-motion-class diagonal; $W=117.50$, $p=0.517$; $BF_{10}=0.42$). Dashed line denotes chance level (3 classes: 33%; 5 classes: 20%). (G) A visualization of the EMG mean relative value profile for the group average (top) and individual participants (bottom) for each gesture across the 8 EMG channels (using the EMG data participants generated to train the classifier). Values reflect the average EMG amplitude at each channel when holding the gesture during calibration. (H) A matrix filled with the euclidean distances between each gesture pair across channels. (I) No differences between trained groups in average euclidean distance across all trained gestures ($W=131.0$, $p=0.346$; $BF_{10}=0.73$). (J) A group comparison between EMG profiles for their first two trained gestures for the arbitrary (1 finger and 2 fingers) and biomimetic users (open and close) revealed more distinct patterns in the biomimetic group ($W=250.0$, $p=0.004$). Circles depict individual subject means (across relevant items). Values indicate group means \pm standard error.

In the Discussion section:

Subsequent analyses found that these early performance differences were likely driven by a combination of cognitive and EMG factors.

...

Furthermore, the arbitrary gestures we tested involved simple finger extensions, and it is likely that more careful curation of control mapping could have produced much greater benefits for the arbitrary control. Indeed, we observed that the initial arbitrary gestures showed more similar EMG patterns than biomimetic gestures. Future studies should develop optimal arbitrary mapping strategies with more distinguishable classification accuracies to biomimetic strategies.

In the Supplementary Results section:

SHAP control speed on Day 1 versus

Supplementary Figure 2. Relationship between Day1 measures. (Left) There was not a significant relationship between SHAP Day 1 control speed and the euclidean distance between the first two trained gestures (used on the task; $r_s=-0.293$, $p=0.08$). (Right) There was a significant relationship between Day 1 control automaticity and SHAP Day 1 control speed ($r_s=-0.478$, $p=0.003$), such that more automatic control reflected faster performance on the SHAP.

In the Methods section on EMG gestural structure:

EMG gestural structure

To quantify the similarity of the EMG patterns for the gestures used in each control strategy, we analyzed the EMG data participants generated when training their classifier. There are multiple EMG features used to make a classification decision, as described in the Calibration protocol section. One of these features is the EMG mean relative value (MRV). To generate a visualization of the EMG profiles, we used the average EMG MRV, computed for each gesture and each channel, during gesture calibration. This data was outputted when each user completed the study. We visualized the MRV for each channel, each gesture, and each subject (Figure 5A). We also averaged the MRV values across subjects of each group to visualize the group-level EMG profile. To quantify the similarity observed in this visualization, we computed the euclidean distance in the EMG MRV values across channels for all gesture pairs.

...

In the section on Statistical analyses:

To investigate whether cognitive (control automaticity) or EMG (euclidean distance between first two trained gestures) factors drove the group differences we observed on the SHAP, we added each measure as a regressor to the SHAP speed scores. The regression model was estimated using ordinary least squares (OLS). We then tested for group differences in the unstandardized residuals of the SHAP day 1 scores, when controlling for each measure.

- 1.3 “Then I have a few remarks on the way results are interpreted to convey the authors’ message. Page 4, Embodiment question. The authors wrote: “Overall, contrary to the common assumptions of biomimetic design, biomimetic control didn’t provide an increased sense of embodiment.” The authors should mitigate this statement, as it could be that the study doesn’t have enough power to show a significant effect. Actually, in terms of ownership, there is a trend towards higher embodiment. Add a “statistically significant” at least in the sentence.”

We agree that the results should be interpreted with caution and in the revisions we’ve added “statistically significant” to the sentence in the revised manuscript.

In the Results section:

“Overall, contrary to the common assumptions of biomimetic design, biomimetic control didn’t provide a **statistically significant** increased sense of embodiment.”

- 1.4 “Then, more in general, the main claim of the paper is that these results should be generalized to the field of prosthetics. I am not sure that this step is so straightforward. There is a big difference between adding and controlling an additional body part and replacing a missing one.

It could be that biomimetic control has much more value for replacing than adding an artificial limb, as tested in a rather specific and ad-hoc context as the one realized here. This needs to be acknowledged, or discussed if the authors do not agree.”

We are happy to expand on this very important consideration. Here, we are asking a fundamental question about control policy of bionic limbs relative to the biological body. We opted to test able-bodied participants because we needed to ensure that all study participants were able to similarly (and accurately) perform the range of gestures tested here. However, it's important to emphasize that for all study purposes, participants were wearing and using the bionic hand for functional replacement of their biological hand. Therefore, the study design is in alignment with prosthesis use, and in fact one could make the argument that biomimetic control is more suitable for our participants' cohort, that switch between their biological hand (outside the study) and the bionic hand throughout the day, relative to amputees' phantom hand. We also wish to emphasize that for prosthesis users that were born without limbs, and never had hand/phantom control, the relationship to substitution is more tenuous, but this limitation equally holds to biomimetic pattern recognition prosthesis control more broadly.

In the revised discussion, we better explain the advantages and disadvantages of our model for clinical application of prosthetic limbs. This includes not only the discussion of replacement versus substitution, but also the relevance of generalization to new control strategies in daily life.

In the Discussion section:

A final limitation is that we tested able-bodied participants using the bionic hand to replace their existing hand instead of limbless prosthesis users, who might incur different benefits and costs to the biomimetic approach. However, considering the primary aim of the study was to mimic bionic limb control to the biological body, we thought it was necessary for us to have direct access to participants' biological limbs to ensure true biomimicry. Moreover, research is now mounting to indicate that amputation might not induce far reaching changes to the motor representation of the missing limb...

1.5 *“Page 10:”For example, learning with a more varied input that increases task difficulty is initially slower, but typically yields better generalization [for a review see (37)]. Accordingly, research has demonstrated that more complex associations end up being learned more robustly and are better retained than easy-to-learn associations (38).” This is a key point to explain the mechanism underlying the effects reported here. It needs to be more precisely discussed and supported either with more straightforward concurrent evidence or better with further analyses of the current data, or at least by proposing further experiments to test this hypothesis.”*

We agree with the reviewer that this is speculative and could have been explained in greater detail within the discussion. In the revised manuscript, we've expanded on this point and proposed the need for future experiments to test this idea.

In the Discussion section:

Why did arbitrary control provide a comparable degree of skill, and even outperform biomimetic

control on generalization? Paradoxically, this could be a result of the cognitively challenging task demands related to adopting atypical gestures for grasp control. While, early in training, *this conflict* would appear to be disadvantageous, *there is ample evidence to suggest that this could actually be the key to improve generalization. The first potential explanation has to do with the increased contextual novelty involved with arbitrary training. Consider, our ability to skillfully use our hands is developed over a lifetime of experience, in which information on motor synergies, object knowledge and action semantics is meticulously weaved to construct efficient motor command pipelines³⁷. Arbitrary users have to learn to associate hand motor commands with entirely unrelated action goals (previously associated with different motor commands). Research has shown that learning with a more varied input that increases task difficulty is initially slower, but typically yields better generalization [for a review see³⁸]. Similarly, stimuli that incorporate a broader range of neural and cognitive processes have been shown to promote increased generalization of learning³⁹. While speculative, it's possible that the arbitrary learning promotes users to develop more general, adaptable representations of action. A second related potential explanation is that arbitrary training requires more cognitive and attentional resources. Research has demonstrated that more complex associations end up being learned more robustly and are better retained, than easy-to-learn associations⁴⁰. Easy-to-learn associations (in our study, biomimetic control) have been shown to rely more heavily on working memory which allows for very fast learning of information that is not durably retained, while learning complex associations, requiring more attention, and relies on reinforcement learning which allows for slow, integrative learning of associations that is robustly stored^{40,41}. Combined with our observation that the arbitrary group had to work with more similar EMG patterns when first learning to use the bionic hand, the increased *cognitive* resources required for arbitrary learning may potentially explain why, *when similar learning was achieved by both groups, the arbitrary control outperformed biomimetic when generalizing to a novel control mapping.**

Reviewer #2:

- 2.1 “Schone and colleagues sought to determine whether biomimetic motor control enhances device embodiment, learning, generalization, and automaticity in anthropomorphic bionic limbs. This is very important to enhance prosthetic functionality and user experience of current and future arm prostheses. The researchers compared biomimetic and non-biomimetic control strategies, with a few well-thought additional controls, by testing able-bodied participants operating a wearable myoelectric bionic hand and providing them an array of user experience tests, falling in two categories: skill tasks and questionnaires. I applaud the authors for the extremely well-crafted experimental design, which is indeed representative of today's commercial technology. The article is very freshly written and engaging. The pop references are on point!”

We want to thank the reviewer for taking the time to review our manuscript and are glad the pop references were appreciated!

2.2 “The figures are perfectly clear, although I would suggest that the authors (as they have done in panel A of Figure 2) would replicate the approach of schematizing the presented test in Fig. 4 and 5.”

Thanks for this suggestion, this has been amended in the revised manuscript for both figures, see below.

In the *Results* section:

“We implemented a dual cognitive-motor task that required participants to perform arithmetic operations while simultaneously using the bionic hand to stack blocks (Supp. Video 1).”

Figure 4. Arbitrary control is less automatic early in training but similar to biomimetic later. (A) Schematic of the dual cognitive-motor task requiring users to simultaneously perform a cognitively demanding arithmetic task aloud (left) while performing a block stacking task (right). Control automaticity was computed by dividing motor performance (number of blocks stacked) when simultaneously performing a counting task by motor performance alone. By this measure, higher values indicate less cognitive encumbrance, or more automatic control. (B) Biomimetic control showed superior performance earlier in training, compared to arbitrary control. However, by the end of training, arbitrary control produced similar performance to that observed in the biomimetic group. (C) All trained participants reported control got easier across the training sessions, regardless of the control strategy. All other annotations are the same as described in Figure 2.

Figure 5. Arbitrary users show increased generalization. (A) During the final study session,

all users swapped to the same novel control mappings (thumbs up = open hand; thumb-pinky pinch = close hand). Performance on **(B)** speed, **(C)** dexterity and **(D)** control difficulty assessments for the generalization session. As a reference for previous performance, the dashed line denotes group mean performance on the last (D4) training session with their original control mappings. Biomimetic users showed significant impairments in comparison to performance during D4. Further, performance in the generalization tasks was similar between the biomimetic group and untrained participants using the bionic hand for the first time. Arbitrary users showed increased generalization such that performance with the new control strategy in the generalization session was similar to performance with the trained strategy on D4. All other annotations are the same as described in Figure 2.

2.3 *“There is a weakness of the article that I wish to discuss, which is also at least in part expressed by the authors as a limitation in the discussion. The experimental design applies to myoelectric prostheses which are slow and actuated by EMG pattern recognition. This is a control bottleneck, that forces the user to 1) substantially anticipate planning knowing they will receive a lagged feedback (= a substantial dose of feed-forward control) and 2) reduce the dimensionality of degrees of freedom in ‘natural’ control to pass through the bottle-neck of a few available patterns. Now, this is exactly how current myoelectric prostheses work. They are mostly slow and bound to a few patterns. This research applies to that egregiously. Yet, the question at the forefront is “Should bionic limb control mimic the human body?”, we are taking a generalization step back to look at defining general laws of human control of bionic limbs. The manuscript is indeed constructed around this very important discussion, but can only address a portion of it, due not to flaws in research design, but to flaws of current technology. I suggest that a more precise definition of the tested hypothesis would be obtained by putting upfront the application to myoelectric pattern recognition prostheses, as a limited proxy of the more general framework.”*

This is a great point, and one that we were very keen to consolidate in our manuscript - when we talk about ‘biomimetic control’ for modern day myoelectric prostheses - what exactly do we mean? In the revised manuscript, we updated the introduction and discussion to really bring this important point to the foreground. Our take home message is that the term biomimetic is relative, and due to its tenuous relationship with natural movement, may not be as promising as the term implies to begin with.

In the *Introduction* section:

Today’s technology is far from Skywalker’s – even the most advanced robotic prosthetics are slow and operated using just a few degrees of freedom. Still, the design of artificial prosthetic hands has steadily innovated appearance and functionality to be increasingly like a biological hand

In the *Discussion* section:

It is important to note that our experimental design may have several potential limitations. First, due to the limited state of modern prosthetic technology (e.g., delay between execution of the motor command to fully engaging motors into the actuated grasp), the biomimetic strategy was

not purely biomimetic (i.e., the same as biological hand control). Additionally, even in the relatively rich realm of pattern recognition, the number of different patterns that are available for control tend to be limited. This means that the natural control that we envisage when considering biomimetic strategies is currently reduced to a limited set of movements. It is possible that as technology progresses to resolve some of these basic bottlenecks, the biomimetic approach will provide more immediate translation. However, by the same token, and given the evidence provided here, it is also possible that as the gap between natural and bionic movement narrows, so does the conflict of the increasingly more subtle differences in motor demands grows. Regardless of these limitations in modern technology, if we consider biomimicry as a spectrum of strategies closer and further away from the biological body, the biomimetic control we tested falls closer to biological control than arbitrary control.

...

So, *given the modern-day technological context of biomimetic control* – practically speaking, how biomimetic should we go? Based on our findings, this depends entirely on the purpose of the prosthesis. If the device is intended for short-term use, simple functionality, or where training opportunities are limited, then biomimetic-inspired control options make a lot of sense. However, if the purpose is to design versatile devices with multiple functions for long-term use, at least with modern EMG pattern recognition technology, non-biomimetic control solutions may provide a useful means to enhance certain aspects of bionic hand motor learning. Further, abandoning the unrealistic ambition of true biomimicry opens up endless possibilities for users and engineers to develop a variety of different control solutions.

2.4 *“As of the higher question, a thought experiment I suggest is the following: what if we were to redo these experiments but using video recognition of hand movements and having to control a virtual hand avatar. Suppose that the control lag is very small, e.g., <50ms. Would you expect not to see a persistent lag of arbitrary strategies with respect to purely biomimetic? Even in light of the current results, I still would expect a performance lag. Perhaps I am wrong, but the question is not resolved and open to discussion and hypotheses. The thought experiment shows that there will be a spectrum of control fidelities to which the general theory applies. Here we are on the lower fidelity side (which are currently state of the art) and the results stand, but better control fidelities may lead to a biomimetic advantage. In summary, I propose 1) to tone down the overall claim to better fit the application sphere of current pattern recognition myoelectric prostheses and 2) to acknowledge that different control fidelities may lead to a different ending balance when comparing the two approaches.”*

While your hypothesis about the potential of biomimetics is intriguing, it is not entirely clear how this would play out in reality, given the current and foreseeable constraints of myoelectric pattern recognition. Specifically - would higher control fidelities possibly lead to more intuitive and efficient use of prosthetics? The general assumption, which you reflect in your comment, is that the closer we can get a prosthesis to mimic a human's movements, the better, assuming all else equal. However, we'd like to add some points for consideration.

First, in our current study design, both groups were affected equally by the control lag (and ensuing low fidelity issues), and if this lag were to decrease significantly, we would expect to see

improvement in both groups. So, there are no immediate reasons why improved fidelity would lead to a biomimetic advantage. Second, as you and us agree, EMG pattern recognition presents significant challenges in this field, due to its context-dependency and difficulty in applying it to diverse real-world situations (i.e. needs to be calibrated in one baseline context and then applied in a very rich space of affordances). Therefore, even with better movement control of the prosthesis in the future, we would still face the challenge of myoelectric pattern recognition, which could still be a limiting factor in achieving high fidelity control. For example, by increasing the similarity between natural and bionic hand control, it's possible that this might lead to an increased conflict, rather than synergy, for the biomimetic group. This is due to the mismatch between the rich natural control our bodies have evolved over millions of years and the classified control that needs to be used for myoelectric pattern recognition. Therefore, high fidelity could paradoxically prove a further disadvantage to the biomimetic strategy.

These considerations are now discussed in our revised discussion, as follows: In the Discussion section:

As device control become more and more biomimetic, but not capable of reaching true biomimicry, it may be creating a more active neurocognitive competition between the priors, sensory predictions, and motor commands for how amputees controlled their pre-existing limb and how they represent and control their prosthetic limb. Therefore, non-biomimetic control solutions may be more beneficial because the motor control plan can be developed from scratch (i.e., independent of pre-existing biological limb control) and therefore can avoid any potential conflict.

...

It is possible that as technology progresses to resolve some of these basic bottlenecks, the biomimetic approach will provide more immediate translation. However, by the same token, and given the evidence provided here, it is also possible that as the gap between natural and bionic movement narrows, so does the conflict of the increasingly more subtle differences in motor demands grows.

Reviewer #3:

- 3.1 *"The authors present a rigorous study addressing an important topic and open question in the field. The manuscript is suitable for publication and would certainly be of interest to the community. That said, the impact of the manuscript is limited in two ways and some claims in the paper should be softened in recognition of these limitations: 1) All tasks can be broken down into a single and simple fundamental challenge that does not address the full span of motor coordination or hand dexterity. In essence, all tasks simply require the user to perform a muscle pattern to activate a grasp, move their forearm, and then perform another muscle pattern to terminate the grasp. Should biomimetic control strategies provide benefits, it's likely these benefits would be observed when tasks are difficult (e.g., with a novel or a complex task). Indeed, data from the manuscript support this idea – as the initial attempts at a task (i.e., when the task is the most novel and complex) have biomimetic control performing better. Ideally, the authors would have explored this idea more fully. A more appropriate test would be to compare biomimetic vs arbitrary control on increasingly difficult tasks that involve more motor*

coordination (e.g., using simultaneous and proportional control of multiple degrees of freedom to complete the same tasks, and more difficult, novel tasks, such as the harder tasks the authors omitted from the SHAP test). That said, I recognize the limitations of EMG control to pursue such an ideal comparison. Instead, I suggest the authors address this limitation in the manuscript and consider exploring this in future work.”

We want to thank the reviewer for taking the time to read through and provide feedback to our manuscript and hope our comments address their concerns. The reviewer suggests that the performance assessments in our study “*simply require the user to perform a muscle pattern to activate a grasp, move their forearm, and then perform another muscle pattern to terminate the grasp.*” Further, the reviewer suggests that if we had used more difficult assessments, they would predict that biomimetic users would consistently outperform arbitrary users. However, we believe the findings we’ve acquired point against this general intuition. The SHAP requires fast muscle contraction, moving the forearm and releasing, very much along the lines of the task demands described by the reviewer. As indicated by the reviewer, as well as in baseline performance, it is the easiest assessment in our battery of tests to complete. Yet, contrary to the reviewer’s prediction – this is where we initially observed group differences in performance. When performing the more difficult assessments, the virtual eggs test (VET) and the gesture switching task, these group differences completely went away. Below, we will break down our tasks to illustrate why the VET and gesture switching assessments are highly difficult and complex, and our considerations for the SHAP. In addition, prompted by the reviewer comments, we dug deeper into the outcome measures of the VET task to see if we can uncover more fine-grained group differences in performance that align with the reviewer’s intuition

Tasks

Virtual Eggs Test

The VET requires proportional control to make sure that the eggs don’t break while they are being transferred. The task is highly difficult; consider that on day 1, the majority of participants (75%) could not transfer a single egg successfully (unbroken) within the 2-minute time period.

Gesture switching task

We implemented two versions of the gesture switching task: 2 gestures (close, pinch) and 3 gestures (close, pinch, tripod). The latter task required users to repeatedly engage and maintain a variety of different muscle contractions consistently for, on average, 3 minutes, while simultaneously quickly grabbing blocks and stacking them into towers. When designing this and all assessments, we consulted with Lynsey Wheelen, an occupational therapist (OTs) and *Director of Remote Training and Occupational Therapy Programs* at Ossur. She (and other OTs we worked with) said that gesture switching at this scale (3 gestures) within the first week of myoelectric training is very uncommon for users, because of the high demand on the muscle and the high likelihood of muscle fatigue. We understand the reviewer’s comment. But, if we would have made this task more difficult, it would not have been feasible for the vast majority of our participants in the 4 days of training.

SHAP

Lastly, the reviewer mentions that it would have been more appropriate to also include the harder tasks of the SHAP test. We did consider using the activities of daily living (ADLs) involved in the SHAP that the reviewer mentions. However, after piloting these tasks when designing the study, we found performance on these tasks was largely dependent on strategy, which is something we were keen to minimize. For this reason, we chose to focus on the abstract object tasks of the SHAP alone because they: (1) are an effective tool to introduce and train new gestures for users, (2) are simple tasks that can be run every single day, and (3) a crude, but effective measure of users ability to quickly engage operate the bionic hand.

In the revised manuscript, we highlight the different motor demands across tasks to better contextualize our results.

Additional analyses

We performed some additional analyses on data not included in the original manuscript to investigate whether there are group differences in other measures from the VET task, beyond just the number of successful eggs transferred. In particular, we looked at the (1) pressure applied to the egg from the bionic hand during the VET task and (2) the percentage of total attempted egg transfers that were successful.

Applied pressure

One thing to consider is that the number of successful eggs transferred is a gross measure of dexterity. A more fine-grained measure is a direct output of the pressure applied by the bionic hand to the eggs. During the VET, we attached pressure sensors to the digit pads of the thumb and index digits to allow us to record the applied pressure (**Revision Figure 3**).

Revision Figure 3. Experimental setup for recording applied pressure during VET. (A) Customized glove fitted to hand with sewn in pressure sensors on the thumb and index digit pads. **(B)** Example of recorded pressure output from each digit when grasping items. **(C)** Example study participant performing the VET with the pressure system.

We quantified the applied pressure during the task for each subject and each day, averaged across both the thumb and index sensors. We observed that all subjects applied less pressure with training (**Revision Figure 4A**; main effect of day: $F_{(3,102)}=5.476$, $p=.002$). Crucially, we did not observe any group differences across sessions (main effect of *group*: $F_{(1,34)}=0.090$, $p=0.765$;

$BF_{\text{incl}}=0.352$). Thus, even with a more fine-grained measure, we still do not see group differences.

Revision Figure 4 (Figure 3B in the revised manuscript). Applied pressure and percentage of attempted egg transfers that were successful during VET. (A) Bionic hand pressure applied to the egg during the VET. Pressure was averaged across both thumb and index sensors. **(B)** The percentage of total attempted egg transfers that were successful/unbroken.

Percentage of total attempted transfers that were successful

In addition to the applied pressure measure, we also wanted to investigate whether there were group differences in the percentage of total attempted egg transfers that were successful. We looked at this measure because it was previously used in a study using the VET (Mastinu et al., 2019; Valle et al., 2018). To quantify this, we went through over 200+ recorded videos of users performing the VET to calculate the total number of attempted eggs transferred (broken and unbroken). We observed that all participants increased the percentage of successful eggs transferred with training (**Revision Figure 4B**; main effect of *day*: $F_{(3,105)} = 8.965$, $p < .001$). Crucially, we did not observe any group differences across days (main effect of group: $F_{(1,35)} = 0.005$, $p = 0.945$, $BF_{\text{incl}} = 0.29$). Combined with the other measures, applied pressure and number of successful eggs transferred, we demonstrate that, across 3 different measures on this task, there were no group differences in dexterity.

Another consideration for the task difficulty of the VET, even after the 4 days of training, only 33% of total egg transfers on average were successful.

In conclusion, we agree with the reviewer that we can't rule out that an advantage for biomimetic would have been observed with different tasks with multiple degrees of freedom. Though, as we've described above, we observed group differences on the easiest task at the beginning of training and not on the more difficult tasks: VET or gesture switching task. As such, we don't think that level of difficulty is a likely explanation for our pattern of results.

We've added these additional analyses to the revised manuscript. In

the Results section:

We next focused on early training performance. To measure control speed, we quantified the ability to operate the hand using completion time on a modified version of the Southampton Hand Assessment Procedure (SHAP; Supp. Video 1). *Note that this task had minimal motor requirements, that is – participants were asked to grasp, transport, and release the grasp of various objects and as such, completion time adequately reflected task performance.* During the first training day, all participants were able to successfully complete the task – *demonstrating the elementary difficulty level* – but biomimetic users were faster than arbitrary users (Figure 3A; D1 performance: $W=85.0$, $p=0.001$). Next, to quantify dexterity, we used the virtual eggs test which measures a users' ability to gently grasp and transport fragile (magnet-fused) blocks (i.e., 'eggs') without breaking them (Supp. Video 1). *This task requires greater motor resources, as demonstrated in previous research^{8,18,19}.* During the first training day, the majority of participants (75%) could not successfully transfer one egg without breaking it within the allocated time – *demonstrating the increased difficulty of this task.* Importantly, there were no group differences (Figure 3B; D1 performance in number of successful eggs transferred: $W=185.50$, $p=0.897$, $BF_{10}=0.34$; *percentage of successful to total eggs transferred: $W=221.50$, $p=0.740$, $BF_{10}=0.32$ and the pressure applied by the bionic hand on egg: $W=202.0$, $p=0.749$, $BF_{10}=0.33$). We also tested the ability of participants to switch between bionic hand gestures. *Fluent switching across multiple gestures is considered an advanced ability for prosthesis users.* To quantify gesture switching, we used completion time on a block stacking task that required participants to successfully grasp and transfer objects using the bionic hand close and pinch gestures, switching back and forth (Supp. Video 1). This ability could only be first tested on Day 2 because participants were only then trained on the second grasping gesture (pinch), thus providing gesture switching functionality. During the first attempt of this task, we observed no group differences in performance (Figure 3C; D2 performance: $W=223.0$, $p=0.537$, $BF_{10}=0.37$). Overall, these findings suggest that biomimetic control affords early training benefits in speed, but not for dexterity and gesture switching.*

...

For the more demanding dexterity task, we did not observe any group differences emerging with training. All trained participants improved in *all dexterity measures* across the training days [Figure 3B; main effect of *day*: number of successful egg transfers: $F_{(3,111)}=14.628$, $p<0.001$; *percentage of total eggs transferred that were successful: $F_{(3,105)}=8.965$, $p<0.001$; applied pressure: $F_{(3,102)}=5.476$, $p=0.002$) and there were no differences between groups (main effect of *group*: number of successful egg transfers: $F_{(1,37)}=0.233$, $p=0.632$, $BF_{incl}=0.27$; *percentage of total eggs transferred that were successful: $F_{(1,35)}=0.005$, $p=0.945$, $BF_{incl}=0.29$; applied pressure: $F_{(1,34)}=0.090$, $p=0.765$, $BF_{incl}=0.35$; for all statistical comparisons see Supp. Table 2). On the last day of training (D4), both groups performed similarly (*number of successful egg transfers: $W=209.0$, $p=0.598$, $BF_{10}=0.32$; percentage of total eggs transferred that were successful: $W=218.0$, $p=0.631$, $BF_{10}=0.32$ and applied pressure: $W=212.0$, $p=0.361$, $BF_{10}=0.33$).***

Figure 3. Bionic hand skill learning on speed, dexterity and gesture switching tasks. (A) Trained participants improved control speed on all **bionic** gestures. For the close gesture, biomimetic control was faster than arbitrary control across training sessions. **(B-C)** Trained participants improved in control dexterity [number of successful/unbroken egg transfers, *percentage of total attempted eggs transferred that were successful, and pressure applied by thumb and index digits during task*] and gesture switching across training sessions, regardless of control strategy. For gesture switching, because participants were trained on new grasping gestures each training day, we used two versions of this task. The 2 gestures version required successfully switching between close and pinch bionic gestures. The 3 gestures version required successfully switching between close, pinch and tripod bionic gestures. No significant differences were found between control strategies. See Supp. Video 1 for examples of all tasks. All other annotations are the same as described in Figure 2.

In the Methods section on Dexterity (Virtual Eggs Test):

Performance was measured as *both* the number of unbroken (successful) eggs *and the number of total attempted eggs transferred within a 2-minute time-period*. *Prior to starting the task, the bionic hand position was pre-set into the open power grip with the thumb manually rotated to be parallel with the other digits*. Participants were allowed to practice transferring one egg.

To quantify pressure applied by the bionic hand, we fitted a custom glove on the bionic hand which included flexiForce sensors [B201-M-8; Tekscan⁶¹] on the thumb and index finger pads. The sensor had a 0.2 mm thickness and a sensing diameter of 9.7 mm. The force sensor was calibrated, based on manufacturer recommendations, by acquiring a linear calibration curve over a range of forces (10, 50, 100, 200, 300, 500 and 1000 g). The sensor has a 0.375-inch sensing area diameter. We used FlexiForce ELF handles (Tekscan) to transfer the pressure data from the sensors to a laptop. Pressure data was recorded from each of these sensors using the ELF System (Tekscan). At the beginning of the VET, we applied a trigger signal to the pressure sensors to use as a start point in the pressure data. For the analysis, the pressure data was averaged across the thumb and index pressure data. Applied pressure was computed as the average pressure during the 2 minute task period.

3.2 *“It would also be appropriate to highlight that the data show biomimetic control consistently outperforming arbitrary control when tasks are difficult (i.e., the first time users completed a task). It remains unclear if arbitrary users could scale performance to biomimetic users when tasks become much more difficult; there may only be so much capacity for an individual to learn a new unintuitive control strategy, such that as the complexity of the hand increases, arbitrary control could lag.”*

The reviewer is correct that the advantage for biomimetic was observed the first-time users completed the SHAP speed test. Though, consider that there were no group differences, when attempting the pinch or tripod versions of the SHAP at all time-points. However, as discussed above, we don't think that task difficulty is the explanation for this advantage. The gesture switching and virtual eggs test are much more difficult assessments, yet biomimetic did not exhibit an advantage over arbitrary at any session. If anything, the data from the present study suggests that as tasks become more difficult, the biomimetic advantage diminishes.

Separate from task difficulty, the reviewer raises another important point: arbitrary strategies potentially are only a viable option when the number of mappings is low. In other words, once the number of mappings increases to a certain point, the cognitive load to learn and consolidate these arbitrary mappings may be too demanding for a user. Though this may not be the case, a relevant example to consider is the popularity of video games. Most modern video game controllers have high control dimensionality to enable the precise control of a virtual avatar/effector. Yet, millions, albeit billions, of individuals successfully learn and master the large number of arbitrary mappings between controller and virtual effector with ease.

Overall, the takeaway here is that the present study finds no evidence to suggest that biomimetic control would provide unique benefits when task difficulty or motor coordination increases. While arbitrary strategies do introduce an initial cognitive hurdle. once overcome, arbitrary control can

perform the same as biomimetic and even outperform biomimetic in generalization.

3.3 “2) In a somewhat similar light, generalization of control strategies should be addressed with a novel task, not a novel arbitrary control strategy. It is somewhat trivial that arbitrary users can generalize better to arbitrary control strategies. For the claims the authors want to make regarding control generalization, they should have performed a novel task instead of a novel control strategy. Further, what are the real-world equivalents of generalization to a novel control strategy? Healthy individuals and prosthesis users do not regularly change the fundamental way they are controlling their hand. More accurately, they encounter new tasks that require new coordinated motor actions that they must learn on the fly. A more appropriate test would be to compare biomimetic and arbitrary control on a novel unseen task. Given this limitation, I suggest the authors soften claims regarding the importance of generalization to different arbitrary control strategy, or explain a real-life use-case and the value of generalization from one arbitrary control strategy to another arbitrary control strategy. Despite the limitations above, the present manuscript makes contributions towards an important question and sets the stage for future work in this area.”

We appreciate the reviewer’s concern about the control generalization finding. Though, we respectfully disagree with the reviewer that our generalization finding is trivial.

Let’s ignore the arbitrary group for a second and just consider the biomimetic and no-training control group. During the generalization session, the no-training control group (users that have never used the device before) perform the same as the biomimetic group in speed ($BF_{10}=0.38$), dexterity ($BF_{10}=0.33$) and their subjective sense of control difficulty ($BF_{10}=0.33$; see **Figure 6**). This is surprising! Even though the biomimetic group spent 4 days (approximately 8 hours) training to operate the device under a variety of conditions and tasks, altering their control strategy completely annihilated their motor performance, such that it is as if they had never trained at all. The fact that biomimetic skill learning is so highly constrained and dependent on their control mapping is what is interesting and hugely important for the future of biomimetic prosthetic control.

We agree that there are different kinds of generalization, such as performing unseen tasks (task generalization) or using a novel control strategy (control generalization). We chose to specifically investigate the latter because it is a fundamental component of real-world myoelectric prosthetic use.

For example, in a recent survey of over 400 prosthetic users, 42% of users reported actively using two or more terminal prosthetic devices (Resnik et al., 2020). Each of these devices – be it a cosmetic, functional hook, simplistic myoelectric, multi-articulating myoelectric – requires entirely different control principles to operate. As such, for users today, an essential component of successful prosthetic use is having flexible, adaptable control or the ability to constantly swap, plug-in and play with a variety of terminal devices without it hindering your performance.

Beyond completely altering control strategies to a new terminal device, control generalization is

also an essential requirement for users to operate even a single myoelectric control strategy for a single terminal device, due to external and internal factors which we will breakdown. Consider that the myoelectric interface – between the surface EMG electrodes and the residual limb – is highly inconsistent throughout the day, due to issues pertaining to socket fit (e.g., increased sweat buildup and/or rain/humidity). More than just changes in these external barriers between sensor and muscle, changes in arm posture during muscle contraction have been shown to impact muscle geometry in different ways (muscle fiber length, diameter, orientation), which in turn can systematically alter the EMG features for a specific muscle (Mesin et al., 2006; Stuttaford et al., 2023). Consequently, a sensor can receive highly variable signals for the same intended contraction. Therefore, users must have flexible, generalizable representations of control gestures, such that their control and motor learning are not strictly dependent on a single strategy, arm posture, time of day, terminal device, etc. Alternatively, users must develop flexible controllers that are adaptable to meet the constant changes in user requirements. It is because of these reasons that our results showing increased control generalization for the arbitrary group are so important for how we think about real world prosthetic use.

We've revised the main text to incorporate both points, see below.

In the Discussion section:

One could ask – is it at all important to generalize across control strategies for real-world myoelectric prosthetic use? Consider that in a recent survey of over 400 prosthetic users, 42% of users reported actively using two or more terminal prosthetic devices⁴⁵. Each of these devices – be it a cosmetic, functional hook, simplistic myoelectric, multi-articulating myoelectric – requires entirely different control principles to operate. As such, for users today, an essential component of successful prosthetic use is having flexible, adaptable control or the ability to constantly swap, plug-in and play with a variety of terminal devices without it hindering your performance. Beyond completely altering control strategies to a new terminal device, control generalization is also an essential requirement for users to operate even a single myoelectric control strategy for a single terminal device, due to external and internal factors. Externally, consider that the myoelectric interface – between the surface EMG electrodes and the residual limb – is highly inconsistent throughout the day⁴⁶, due to issues pertaining to socket fit, which can often be exacerbated by sweat buildup and/or rain/humidity. Internally, changes in arm posture during muscle contraction have been shown to impact muscle geometry in different ways (muscle fiber length, diameter, orientation), which in turn can systematically alter the EMG features for a specific muscle^{47,48}. Consequently, a sensor can receive highly variable signals for the same intended contraction. Therefore, users must have flexible, generalizable representations of control gestures, such that their control and motor learning are not strictly dependent on a single strategy, arm posture, time of day, terminal device, task, etc. This is why the increased control generalization for the arbitrary group is so important for how we think about real world prosthetic use and the future of biomimetic prosthetics.

...

A third limitation is that perhaps arbitrary users showed greater generalization than biomimetic users simply because the generalization session control mappings are essentially just a different arbitrary strategy. However, it's important to consider just the biomimetic and no-

training control groups. During the generalization session, the no-training control group (users that have never used the device before) perform the same as the biomimetic group in speed ($BF_{10}=0.38$), dexterity ($BF_{10}=0.33$) and subjective sense of control difficulty ($BF_{10}=0.33$). Even though the biomimetic group spent 4 days (approximately 8+ hours) training to operate the device under a variety of conditions and tasks, altering their control strategy completely annihilated their motor performance, such that it is as if they had never trained at all. The fact that biomimetic skill learning was so highly constrained to their control mapping is what is so interesting.

3.4 *“Some additional comments are below: Use of 1 – 4 gestures is not “as far away from biomimetic” as possible. It’s logical and intuitive in that it’s the order the gestures were introduced.”*

We completely agree with the reviewer that the arbitrary gestures used in the present study are not the most non-biomimetic choice possible. We argue for a biomimetic to arbitrary spectrum. Within that spectrum, our biomimetic control is not pure biomimetic (based on limitations with modern myoelectric control). Similarly, our arbitrary control was not designed to be the most arbitrary. The arbitrary gestures we chose crucially have no functional role in action or object interaction. As such, the biomimetic control we tested falls closer to biological control than the arbitrary control. We hoped this was clear in what we wrote in the discussion section:

In the *Introduction* section:

*As a striking alternative to biomimetic control, we incorporated an arbitrary (non-biomimetic) control strategy. Based on the neurocognitive assumptions underlying biomimetic design, this strategy should provide no direct benefits for the user. The primary rationale of the arbitrary strategy is to provide a contrast to biomimetic. **When choosing arbitrary gestures, we prioritized gestures not involved in typical object interaction, but are otherwise easy to instruct, memorize, execute and replicate (e.g., 1 finger to 4 fingers). The main idea is that this control strategy is moving away from natural movement and prioritizing considerations other than biomimetics.***

In the *Discussion* section:

...if we consider biomimicry as a spectrum of strategies closer and further away from the biological body, the biomimetic control we tested falls closer to biological control than arbitrary control.

3.5 *“Not included is preference among control strategies. Patients report wanting bionic hands that are visually similar to human hands. They may also simply prefer biomimetic despite it not necessarily providing optimal control (similar to how patients use non-function cosmetic prostheses).”*

We agree with the reviewer that users may ask for biomimetic simply because of the low cognitive demand. Which is why we argue, not for constraining users to a single strategy, but rather for educating users on the spectrum of strategies available to them. The purpose of this study is to understand the neurocognitive opportunities and limitations of different strategies

across this spectrum to help inform the choices users make.

In the Discussion section:

We suggest that engineers and prosthetists involved in the commercial and clinical delivery of this technology should prioritize flexibility – educating users on the spectrum of biomimetic-to-arbitrary control strategies available to them such that personalized, user-specific control strategies can be selected based on individual user requirements. In our experience, when users are educated with the knowledge and confidence that multiple control approaches are possible, there is a higher likelihood that devices will meet user expectations and requirements. Personalized control strategies will help to propel the industry closer to the actual goal: more satisfied prosthesis users.

3.6 “No information is provided on the proportional control needed for the VET. Please provide methodology for how the myoelectric classifier provides proportional control. Are users controlling hand position directly? Hand velocity?”

Prior to starting the VET, the hand position was pre-set into the open cylindrical grip with the thumb manually rotated to be parallel with the other digits. Users control the velocity at which the hand closes; the velocity being proportional to the EMG activity (for details on this methodology, Scheme & Englehart, 2011).

We’ve updated the methods section to clarify this. In

the Methods, Dexterity (Virtual Eggs Test):

Prior to starting the task, the bionic hand position was pre-set into the open power grip with the thumb manually rotated to be parallel with the other digits.

In the Methods, Calibration protocol section:

Bionic hand velocity control was proportional to the EMG activity.

3.7 “How do the values reported here compare to other uses of the VET, SHAP and mode-switching tasks? Are the users in this study performing much worse or much better overall? This comparison can help assess if performance was simply due to a task being too easy or too difficult.”

Generally, we feel it is not straightforward to directly compare performance on these assessments across studies, due to differences in terminal devices, EMG controllers, able-bodied vs. amputees, previous myoelectric experience etc. All of these factors will impact how users perform on these assessments. Regardless, we attempted to do comparisons to other datasets that are the most relevant.

SHAP

For the SHAP, most previous studies report SHAP scores as a single compound measure that incorporates performance across multiple different tasks. This is contrary to the approach we took in the present study which involved selecting specific tasks within the SHAP that most directly reflected performance on a specific skill and reporting the raw results for each of these

tasks. Regardless, there is a report showing speed performance for individual SHAP tasks for 27 prosthesis users (Burgerhof et al., 2017). They reported the following median transfer times: light power grip: 3.75 seconds, light pinch grip: 4.63 s, and light tripod grip: 4.94 s. We next adjust these measures to our study design involving 20 transfers, therefore the equivalent median transfer times from this report would be: light power (close) grip: 75 s (i.e., 3.75s X 20), light pinch grip: 92.6 s, and light tripod grip: 98.8 s.

In the present study, Day 1 median RT for light power grip: biomimetic, 63.56 s; arbitrary: 97.47 s, light pinch grip: biomimetic, 70.72 s; arbitrary: 60.08 s, light tripod grip: biomimetic: 72.96 s, arbitrary: 84.15 s.

	Power Grip (median time: seconds)	Pinch Grip	Tripod Grip
Biomimetic Users	63.5	70.7	72.9
Arbitrary Users	97.4	60.0	84.1
Burgerhof et al., 2017	75.0	92.6	98.8

Overall, our reaction-times are slightly faster than the report above (unsurprising since our users are able-bodied), though arbitrary Day 1 scores are on average slightly slower than the amputees scores. In summary, for this task, it's fair to say it's a simple, effective measure of control speed for a variety of gestures and that our values are somewhat similar to previous reports.

VET

For the virtual eggs test, very few studies have actually used it, as it has not yet been clinically validated (Clemente et al., 2016; Mastinu et al., 2019; Valle et al., 2018). Further, the studies that have used it have implemented very different fragile objects in the assessment [e.g., from using paper boxes held together with spaghetti: Clemente et al., 2016, to, what we used in the present study, magnetic fused 3D printed blocks: Mastinu et al., 2019], so we don't think it's a fair comparison.

Regardless, Valle et al., 2018 performed the VET in a small sample of amputees (<5) using a custom-terminal hand and a break threshold of their fragile objects at 1.2 N. The study was specifically looking at the effect of sensory feedback on task performance. They observed that the percentage of total attempted eggs transferred successfully was approximately 45% without sensory feedback and 70% with sensory feedback. In the present study, on Day 1, on average for all subjects, approximately 15% of total eggs were transferred successfully and, on Day 4, approximately 33% were successful. This is most likely due to them using a custom-terminal hand, while we used a commercial device (the Ossur iLIMB quantum), however it could also reflect the training demands of proportional EMG control (Valle's participants were expert prosthesis users). In another study, Mastinu and colleagues tested 3 transhumeral amputees performing the VET over 5 sessions looking at break thresholds of 6N

(approximately what we used in the present study) and 12N (Mastinu et al., 2019). For the 6N break threshold, they observed that the first amputee (S1) improved over 5 sessions from approximately 0% (session 1) to 60% (session 5) of total eggs transferred successfully. The second amputee (S2) got worse over the sessions from 70% (session 1) to 50% (session 5) successful transfers. Lastly, the third amputee (S3) got slightly worse from 100% to 80% (session 5). Overall, the main takeaway from these study comparisons is that our participants found the task harder, as such this should be considered in relation to the reviewer's previous comment about task difficulty: **reviewer response 3.1**.

Gesture switching

Our gesture switching tasks were unique and specifically designed to test gesture switching for the Coapt - iLIMB control configuration. As such, it's difficult to make a valid comparison to any other studies.

3.8 "What is the ratio of object weight to break force used in the VET test? How does this compare to prior uses of fragile object tasks? Is this an easy version of a fragile object test or a difficult version?"

As detailed in our previous response, our participants found the task to be difficult, in comparison to previous studies. We agree with the reviewer that future clinical efforts that seek to validate the use of the VET as a clinical outcome assessment for prosthetic control and learning should consider how object weight contributes to different breaking forces and how the VET compares to other fragile object tests.

Reviewer #4:

4.1 "This is a well written paper describing an interesting study and experimental paradigm. I have just a few comments. First, the lack of 'generalizability' does not necessarily seem surprising as in one case, users are trained on an arbitrary mapping and the other case a biomimetic mapping. In the generalization, they are trained in a different arbitrary mapping – which would seem to bias learning toward the first group. It might be interesting instead to swap control approaches at this phase (i.e., arbitrary then does biomimetic). I don't think this needs to be done at this stage, but the point could be better addressed in the discussion."

We appreciate the reviewer's concern about the control generalization finding. Though, we respectfully disagree with the reviewer that our generalization finding is unsurprising.

Let's ignore the arbitrary group for a second and just consider the biomimetic and no-training control group. During the generalization session, the no-training control group (users that have never used the device before) perform the same as the biomimetic group in speed ($BF_{10}=0.38$), dexterity ($BF_{10}=0.33$) and their subjective sense of control difficulty ($BF_{10}=0.33$; see **Figure 5**). This is surprising! Even though the biomimetic group spent 4 days (approximately 8 hours) training to operate the device under a variety of conditions and tasks, altering their control

strategy completely annihilated their motor performance, such that it is as if they had never trained at all. The fact that biomimetic skill learning is so highly constrained and dependent on their control mapping is what is interesting and hugely important for the future of biomimetic prosthesis control.

The complete generalization we observed for arbitrary trained users is an important consideration for how we envision the future of prosthetic control. Indeed, the ability to generalize control is an essential feature for successful control of myoelectric prosthetics. For example, in a recent survey of over 400 prosthetic users, 42% of users reported actively using two or more terminal prosthetic devices (Resnik et al., 2020). Each of these devices – be it a cosmetic, functional hook, simplistic myoelectric, multi-articulating myoelectric – requires entirely different control principles to operate. As such, for users today, an essential component of successful prosthetic use is having flexible, adaptable control or the ability to constantly swap, plug-in and play with a variety of terminal devices without it hindering your performance.

Beyond completely altering control strategies to a new terminal device, control generalization is also an essential requirement for users to operate even a single myoelectric control strategy for a single terminal device, due to external and internal factors which we will breakdown. Consider that the myoelectric interface – between the surface EMG electrodes and the residual limb – is highly inconsistent throughout the day, due to issues pertaining to socket fit (e.g. increased sweat buildup and/or rain/humidity). More than just changes in these external barriers between sensor and muscle, changes in arm posture during muscle contraction have been shown to impact muscle geometry in different ways (muscle fiber length, diameter, orientation), which in turn can systematically alter the EMG features for a specific muscle (Mesin et al., 2006; Stuttford et al., 2023). Consequently, a sensor can receive highly variable signals for the same intended contraction. Therefore, users must have flexible, generalizable representations of control gestures, such that their control and motor learning are not strictly dependent on a single strategy, arm posture, time of day, terminal device, task, etc. Alternatively, users must develop flexible controllers that are adaptable to meet the constant changes in user requirements. It is because of these reasons that our results showing increased control generalization for the arbitrary group are so important for how we think about real world prosthetic use.

In the revised manuscript, we've added these considerations to the discussion.

In the Discussion section:

One could ask – is it at all important to generalize across control strategies for real-world myoelectric prosthetic use? Consider that in a recent survey of over 400 prosthetic users, 42% of users reported actively using two or more terminal prosthetic devices⁴⁵. Each of these devices – be it a cosmetic, functional hook, simplistic myoelectric, multi-articulating myoelectric – requires entirely different control principles to operate. As such, for users today, an essential component of successful prosthetic use is having flexible, adaptable control or the ability to constantly swap, plug-in and play with a variety of terminal devices without it hindering your performance. Beyond completely altering control strategies to a new terminal device, control generalization is also an essential requirement for users to operate even a single myoelectric control strategy for a single

terminal device, due to external and internal factors. Externally, consider that the myoelectric interface – between the surface EMG electrodes and the residual limb – is highly inconsistent throughout the day⁴⁶, due to issues pertaining to socket fit, which can often be exacerbated by sweat buildup and/or rain/humidity. Internally, changes in arm posture during muscle contraction have been shown to impact muscle geometry in different ways (muscle fiber length, diameter, orientation), which in turn can systematically alter the EMG features for a specific muscle^{47,48}. Consequently, a sensor can receive highly variable signals for the same intended contraction. Therefore, users must have flexible, generalizable representations of control gestures, such that their control and motor learning are not strictly dependent on a single strategy, arm posture, time of day, terminal device, task, etc. This is why the increased control generalization for the arbitrary group is so important for how we think about real world prosthetic use and the future of biomimetic prosthetics.

...

A third limitation is that perhaps arbitrary users showed greater generalization than biomimetic users simply because the generalization session control mappings are essentially just a different arbitrary strategy. However, it's important to consider just the biomimetic and no-training control groups. During the generalization session, the no-training control group (users that have never used the device before) perform the same as the biomimetic group in speed ($BF_{10}=0.38$), dexterity ($BF_{10}=0.33$) and subjective sense of control difficulty ($BF_{10}=0.33$). Even though the biomimetic group spent 4 days (approximately 8+ hours) training to operate the device under a variety of conditions and tasks, altering their control strategy completely annihilated their motor performance, such that it is as if they had never trained at all. The fact that biomimetic skill learning was so highly constrained to their control mapping is what is so interesting.

Minor comments:

4.2 “Biomimetic advantage section, paragraph 3 – ‘no significant differences’ should say ‘in’ not ‘on’ (line numbers would help review)”

This has been amended.

4.3 “Subsequent section – suggest rephrasing to ‘more affected by the cognitive task’ as cognitively impaired implies something different”

Agreed, this has been amended.

4.4 “Remove ref 42 as paper is not yet submitted”

This has been amended.

Revision Response References

Burgerhof, J. G. M., Vasluian, E., Dijkstra, P. U., Bongers, R. M., & van der Sluis, C. K. (2017). The Southampton Hand Assessment Procedure revisited: A transparent linear scoring system, applied to data of experienced prosthetic users. *Journal of Hand Therapy: Official Journal of the American Society of Hand Therapists*, 30(1), 49–57. <https://doi.org/10.1016/j.jht.2016.05.001>

Clemente, F., D'Alonzo, M., Controzzi, M., Edin, B. B., & Cipriani, C. (2016). Non-Invasive, Temporally Discrete Feedback of Object Contact and Release Improves Grasp Control of Closed-Loop Myoelectric Transradial Prostheses. *IEEE Transactions on Neural Systems and Rehabilitation Engineering: A Publication of the IEEE Engineering in Medicine and Biology Society*, 24(12), 1314–1322. <https://doi.org/10.1109/TNSRE.2015.2500586>

Mastinu, E., Clemente, F., Sassu, P., Aszmann, O., Brånemark, R., Håkansson, B., Controzzi, M., Cipriani, C., & Ortiz-Catalan, M. (2019). Grip control and motor coordination with implanted and surface electrodes while grasping with an osseointegrated prosthetic hand. *Journal of NeuroEngineering and Rehabilitation*, 16(1), 49. <https://doi.org/10.1186/s12984-019-0511-2>

Mesin, L., Joubert, M., Hanekom, T., Merletti, R., & Farina, D. (2006). A finite element model for describing the effect of muscle shortening on surface EMG. *IEEE Transactions on Biomedical Engineering*, 53(4), 593–600. <https://doi.org/10.1109/TBME.2006.870256>

Resnik, L., Borgia, M., Heinemann, A. W., & Clark, M. A. (2020). Prosthesis satisfaction in a national sample of Veterans with upper limb amputation. *Prosthetics and Orthotics International*, 44(2), 81–91. <https://doi.org/10.1177/0309364619895201>

Scheme, E., & Englehart, K. (2011). Electromyogram pattern recognition for control of powered upper-limb prostheses: State of the art and challenges for clinical use. *Journal of Rehabilitation Research and Development*, 48(6), 643–659. <https://doi.org/10.1682/jrrd.2010.09.0177>

Stuttaford, S. A., Dyson, M., Nazarpour, K., & Dupan, S. S. G. (2023). *Reducing Motor Variability Enhances Myoelectric Control Robustness Across Limb Positions* (p. 2023.05.05.539580). bioRxiv. <https://doi.org/10.1101/2023.05.05.539580>

Valle, G., Mazzoni, A., Iberite, F., D'Anna, E., Strauss, I., Granata, G., Controzzi, M., Clemente, F., Rognini, G., Cipriani, C., Stieglitz, T., Petrini, F. M., Rossini, P. M., & Micera, S. (2018). Biomimetic Intraneural Sensory Feedback Enhances Sensation Naturalness, Tactile Sensitivity, and Manual Dexterity in a Bidirectional Prosthesis. *Neuron*, 100(1), 37–45.e7. <https://doi.org/10.1016/j.neuron.2018.08.033>

Decision Letter, first revision:

20th November 2023

Dear Dr. Schone,

Thank you for your patience as we've prepared the guidelines for final submission of your Nature Human Behaviour manuscript, "Should bionic limb control mimic the human body? Impact of control strategy on bionic hand skill learning" (NATHUMBEHAV-23020522A). Please carefully follow the step-by-step instructions provided in the attached file, and add a response in each row of the table to indicate the changes that you have made. Please also address the additional marked-up edits we have proposed within the reporting summary. Ensuring that each point is addressed will help to ensure that your revised manuscript can be swiftly handed over to our production team.

We would hope to receive your revised paper, with all of the requested files and forms within two-three weeks. Please get in contact with us if you anticipate delays.

If you have not done so already, please alert us to any related manuscripts from your group that are under consideration or in press at other journals, or are being written up for submission to other journals (see: [https://www.nature.com/nature-research/editorial-policies/plagiarism](https://www.nature.com/nature-research/editorial-policies/plagiarism#policy-on-duplicate-publication) #policy-on-duplicate-publication for details).

Nature Human Behaviour offers a Transparent Peer Review option for new original research manuscripts submitted after December 1st, 2019. As part of this initiative, we encourage our authors to support increased transparency into the peer review process by agreeing to have the reviewer comments, author rebuttal letters, and editorial decision letters published as a Supplementary item. When you submit your final files please clearly state in your cover letter whether or not you would like to participate in this initiative. Please note that failure to state your preference will result in delays in accepting your manuscript for publication.

In recognition of the time and expertise our reviewers provide to Nature Human Behaviour's editorial process, we would like to formally acknowledge their contribution to the external peer review of your manuscript entitled "Should bionic limb control mimic the human body? Impact of control strategy on bionic hand skill learning". For those reviewers who give their assent, we will be publishing their names alongside the published article.

Cover suggestions

We welcome submissions of artwork for consideration for our cover. For more information, please see our https://www.nature.com/documents/Nature_covers_author_guide.pdf target="new"> guide for cover artwork.

ORCID

Non-corresponding authors do not have to link their ORCIDs but are encouraged to do so. Please note that it will not be possible to add/modify ORCIDs at proof. Thus, please let your co-authors know that if they wish to have their ORCID added to the paper they must follow the procedure described in the following link prior to acceptance: <https://www.springernature.com/gp/researchers/orcid/orcid-for-nature-research>

Nature Human Behaviour has now transitioned to a unified Rights Collection system which will allow our Author Services team to quickly and easily collect the rights and permissions required to publish your work. Approximately 10 days after your paper is formally accepted, you will receive an email in providing you with a link to complete the grant of rights. If your paper is eligible for Open Access, our Author Services team will also be in touch regarding any additional information that may be required to arrange payment for your article.

Please note that *Nature Human Behaviour* is a Transformative Journal (TJ). Authors may publish their research with us through the traditional subscription access route or make their paper immediately open access through payment of an article-processing charge (APC). Authors will not be required to make a final decision about access to their article until it has been accepted. Find out more about Transformative Journals

[REDACTED]

Best regards,
Alex McKay
Editorial Assistant
Nature Human Behaviour

On behalf of

Giacomo Ariani
Editor
Nature Human Behaviour

Reviewer #1:

Remarks to the Author:

The authors provided detailed answers to the points I raised and revised the manuscript to address all my specific concerns.

Reviewer #2:

Remarks to the Author:

I'd like to thank the authors for taking into account my comments.

I find convincing the statement that a key takeaway is that the term "biomimetic" is relative.

While I disagree with the authors' hypothesis that scaling control quality beyond pattern recognition could lead to increased conflict rather than synergy in biomimetic strategies, I understand their perspective.

I see the evolutionary argument regarding the overwhelming specificity of hand control. However, I still believe that further refinement in myoelectric control could enhance intuitiveness. While new specialized control conflicts may arise, the more significant issues are progressively resolved. Nevertheless, it's just my opinion. I respect the authors' point of view and believe they should present their argument "as is" in the discussion.

I would still like to see the term 'pattern recognition' highlighted in either the title or abstract, as I consider it a significant disclaimer of the framework of this study.

I don't have further comments. Thank you to the authors.

Reviewer #3:

Remarks to the Author:

My initial review provided 3 major points, broken out by the authors as 3.1, 3.2, and 3.3. I also raised several minor points that the authors adequately addressed.

I still disagree somewhat with the authors regarding point 3.1. I understand the tasks presented here were difficult for the users to achieve given the modern capabilities of EMG control. However, more conceptually, I believe the findings are limited in that they are all centered around discrete and sequential classification. One could argue by definition that nothing within the realm of sequential classification will ever be truly biomimetic. I recognize it's outside the scope of the present study, but I suggest the authors consider how this work might translate to the alternative control domain involving simultaneous and proportional regression of individual degrees of freedom, which are then combined to enable more complex and novel tasks.

I agree with the authors' response to 3.2. I particularly resonated with the video game controller analogy. I suggest the authors consider adding this to the manuscript text.

I disagree with the authors' response to 3.3. I note that reviewer 4 also brought up this point (4.1). I again recommend the authors tone down the manuscripts broad claims and instead focus specifically on the field of pattern recognition EMG. This is also captured by reviewer 2 (2.4), "tone down the overall claim to better fit the application sphere of current pattern recognition myoelectric prostheses." Throughout the manuscript, rather than adding more speculative discussion, I suggest keeping the focus more concise. For this particular point (3.3), I would rather omit the added speculation and simply acknowledge the limitation. The added text has been counterproductive; it has only added additional broad sweeping claims that extrapolate too far beyond the scope of the present manuscript. As written, it appears the authors are now also implying arbitrary control will enable better performance across different prostheses (unproven)

and provide more robust control to changes in the EMG signals (unproven) and the EMG interface (unproven).

Reviewer #4:

Remarks to the Author:

The authors have done significant work in addressing prior concerns. The revised paper provides better context around the findings. I have no additional comments

Author Rebuttal, first revision:

REVIEWER COMMENTS:

Reviewer #1:

1.1 *“The authors provided detailed answers to the points I raised and revised the manuscript to address all my specific concerns.”*

We want to thank the reviewer for their feedback, which we believe greatly improved the manuscript.

Reviewer #2:

2.1 *“I'd like to thank the authors for taking into account my comments. I find convincing the statement that a key takeaway is that the term "biomimetic" is relative. While I disagree with the authors' hypothesis that scaling control quality beyond pattern recognition could lead to increased conflict rather than synergy in biomimetic strategies, I understand their perspective. I see the evolutionary argument regarding the overwhelming specificity of hand control. However, I still believe that further refinement in myoelectric control could enhance intuitiveness. While new specialized control conflicts may arise, the more significant issues are progressively resolved. Nevertheless, it's just my opinion. I respect the authors' point of view and believe they should present their argument "as is" in the discussion.”*

We also respect the reviewer's perspective and have made sure the discussion is well-balanced on the matter.

2.2 *“I would still like to see the term 'pattern recognition' highlighted in either the title or abstract, as I consider it a significant disclaimer of the framework of this study. I don't have further comments. Thank you to the authors.”*

We've added the term 'pattern recognition' to the abstract.

In the Abstract:

“To test this, we compared biomimetic and non-biomimetic control strategies for able-bodied participants when learning to control a wearable myoelectric bionic hand operated by an 8-channel EMG pattern recognition system.”

Reviewer #3:

3.1 *“My initial review provided 3 major points, broken out by the authors as 3.1, 3.2, and 3.3. I also raised several minor points that the authors adequately addressed. I still disagree somewhat with the authors regarding point 3.1. I understand the tasks presented here were difficult for the users to achieve given the modern capabilities of EMG control. However, more conceptually, I believe the findings are limited in that they are all centered around discrete and sequential classification. One could argue by definition that nothing within the realm of sequential classification will ever be truly biomimetic. I recognize it's outside the scope of the present study, but I suggest the authors consider how this work might translate to the alternative control domain involving simultaneous and proportional regression of individual degrees of freedom, which are then combined to enable more complex and novel tasks.”*

The important consideration the reviewer highlights is that present technology in the manuscript places limitations on the extent to which we can explore true biomimetic control. This is of course something we are happy to acknowledge, as reflected in the revised Abstract, Introduction and Discussion sections. However, this does not mean we cannot test the relative differences between control strategies that are more biomimetic to ones that are arbitrary, even through the lens of EMG pattern recognition. This is now articulated more carefully in the revised Introduction and Discussion.

In the Abstract:

“To test this, we compared biomimetic and non-biomimetic control strategies for able-bodied participants when learning to control a wearable myoelectric bionic hand operated by an 8-channel EMG pattern recognition system.”

In the Introduction section:

*“Here, we compared biomimetic and non-biomimetic motor control strategies directly while participants learned to control a bionic hand, **operated by an 8-channel EMG***

pattern recognition system (i.e., Coapt; the most advanced commercially available system for controlling myoelectric bionic limbs). As a striking alternative to biomimetic control (as implemented for existing myoelectric technology), we incorporated an arbitrary (non-biomimetic) control strategy.”

In the Discussion section:

*“Additionally, even in the relatively rich realm of pattern recognition, the number of different patterns that are available for control tend to be limited, i.e., **restricted to discrete and sequential classification**. This means that the natural control that we envisage when considering biomimetic strategies is currently reduced to a limited set of movements. It is possible that as technology progresses to resolve some of these basic bottlenecks, the biomimetic approach will provide more immediate translation.*

...if we consider biomimicry as a spectrum of strategies closer and further away from the biological body, the biomimetic control we tested falls closer to biological control than arbitrary control.”

As recognized by the reviewer, we feel discussing this more than we already are in the current manuscript is beyond the scope of the present study. Though, we strongly agree with the reviewer of the importance for future studies to develop investigations that attempt to explore the neurocognitive advantages and limitations of more biomimetic control strategies.

3.2 “I agree with the authors' response to 3.2. I particularly resonated with the video game controller analogy. I suggest the authors consider adding this to the manuscript text.”

We appreciate that the reviewer found the analogy helpful and have added it to the Discussion section.

In the Discussion section:

“Though, this raises an important consideration for the scalability of arbitrary control in future bionics, namely that as the number of arbitrary mappings increases the cognitive load to learn and consolidate these mappings could become too demanding for a user. However, a relevant example to consider is the popularity of video games that require complex control. Most modern video game controllers have high control dimensionality to enable the precise control of a virtual avatar/effector. Yet, millions, albeit billions, of individuals successfully learn and master the large number of arbitrary mappings between controller and virtual effector with ease.”

3.3 *"I disagree with the authors' response to 3.3. I note that reviewer 4 also brought up this point (4.1). I again recommend the authors tone down the manuscripts broad claims and instead focus specifically on the field of pattern recognition EMG. This is also captured by reviewer 2 (2.4), "tone down the overall claim to better fit the application sphere of current pattern recognition myoelectric prostheses." Throughout the manuscript, rather than adding more speculative discussion, I suggest keeping the focus more concise. For this particular point (3.3), I would rather omit the added speculation and simply acknowledge the limitation. The added text has been counterproductive; it has only added additional broad sweeping claims that extrapolate too far beyond the scope of the present manuscript. As written, it appears the authors are now also implying arbitrary control will enable better performance across different prostheses (unproven) and provide more robust control to changes in the EMG signals (unproven) and the EMG interface (unproven)."*

The reviewer makes two points above that we want to respond to: (1) the manuscript's claims are too broad, and (2) the additional text surrounding one of the limitations discussed was too speculative.

Let's address each of these points in turn. First, in the manuscript's Introduction and Discussion sections, we discuss how EMG prosthetics are a useful model for examining the role of biomimicry in assistive technologies because it allowed us to have direct access to both biological and bionic limb control, all within the same paradigm. While we are very happy to acknowledge that our findings are confounded by our technology of choice, they nevertheless provide a valuable model for motor control along the spectrum of biomimicry. There are other models we could have used (e.g., BCI controlled robotic limbs, physical controllers of virtual effectors etc), each though with their own inherent strengths and limitations. Simply because our model was pattern recognition EMG, does not mean the implications of our findings do not extend to other technologies. Indeed, how to map the high dimensionality of biological limb control to operate bionic limbs is a necessary design consideration impacting a variety of technologies throughout the neurotechnology and rehabilitation realm, such as brain-computer interfaces, teleoperated surgical robots, and human augmentation (more generally). Which is why, we agree with this reviewer's initial response to the manuscript that *"the present manuscript makes contributions towards an important question and sets the stage for future work in this area."* The larger question the reviewer highlights and the future work remaining for the field extends well beyond EMG prosthetics but rather what is the role of biomimicry (more generally) in human-machine interfaces.

The reviewer's second point is that they would: "*rather omit the added speculation and simply acknowledge the limitation.*" Of the two text additions we made in the 3.3 reviewer response, only one was in reference to a limitation, so we believe this is the added text the reviewer is referring to here, see below:

Previously revised text in the *Discussion* section:

"A third limitation is that perhaps arbitrary users showed greater generalization than biomimetic users simply because the generalization session control mappings are essentially just a different arbitrary strategy. However, it's important to consider just the biomimetic and no-training control groups. During the generalization session, the no-training control group (users that have never used the device before) perform the same as the biomimetic group in speed ($BF_{10}=0.38$), dexterity ($BF_{10}=0.33$) and subjective sense of control difficulty ($BF_{10}=0.33$). Even though the biomimetic group spent 4 days (approximately 8+ hours) training to operate the device under a variety of conditions and tasks, altering their control strategy completely annihilated their motor performance, such that it is as if they had never trained at all. The fact that biomimetic skill learning was so highly constrained to their control mapping is what is so interesting."

In the previous revision, we agreed with the reviewer about this limitation, which is why we acknowledge it in the Discussion section. Though, we disagree that the added text following this limitation is at all speculative (i.e., based on conjecture rather than knowledge). In the added text, we are simply recapping our findings as they relate to the limitation. From our perspective, when interpreting the generalization session data, there are two considerations: (1) why did the arbitrary group maintain their skill in the generalization session, and (2) why did the biomimetic group show no advantage relative to first time users. The added text is in reference to the latter consideration, that is, that despite 4 days of training, biomimetic users perform the same as first-time users. This is important context surrounding the presented limitation. Regardless of whether the arbitrary generalization result is strictly due to a bias (e.g., generalization session included different arbitrary gestures), wouldn't we still predict that biomimetic users (based on their extensive training) would perform better than first-time users (e.g. untrained group)? Considering biomimetic users perform the same as first-time users during the generalization session, this suggests that an arbitrary bias alone is not sufficient for explaining our findings. The added text incorporates this consideration, such that the limitation is presented in a well-balanced manner to the reader. Overall, we do not agree that the added text is speculative, but rather helpful context for considering the potential limitation. Nevertheless, we have made light edits to this text to better flesh out our argument, and we leave it to the Editor's discretion to decide whether this paragraph should stay or be removed.

Revised text in the *Discussion* section:

“A third limitation is that perhaps arbitrary users showed greater generalization than biomimetic users simply because the generalization session control mappings are essentially just a different arbitrary strategy. **However, regardless of why the arbitrary users showed an increased generalization, why did the biomimetic users show no advantage relative to first time users in the generalization session?** Consider that, during the generalization session, the no-training control group (users that have never used the device before) performed the same as the biomimetic group in speed ($BF_{10}=0.38$), dexterity ($BF_{10}=0.33$) and subjective sense of control difficulty ($BF_{10}=0.33$).

Despite the biomimetic group spending 4 days (approximately 8+ hours) training to operate the device under a variety of conditions and tasks, altering their control strategy completely annihilated their motor performance, such that it is as if they had never trained at all. **While we cannot disregard the potential bias the choice of generalization gestures had for arbitrary users, the fact that biomimetic skill learning was so highly constrained to their control mapping is what is so interesting.** ”

Reviewer #4:

4.1 “The authors have done significant work in addressing prior concerns. The revised paper provides better context around the findings. I have no additional comments”

We want to thank the reviewer for all of their feedback!

Final Decision Letter:

Dear Dr Schone,

We are pleased to inform you that your Article "Biomimetic versus arbitrary motor control strategies for bionic hand skill learning", has now been accepted for publication in Nature Human Behaviour.

Please note that *Nature Human Behaviour* is a Transformative Journal (TJ). Authors may publish their research with us through the traditional subscription access route or make their paper immediately open access through payment of an article-processing charge (APC). Authors will not be required to make a final decision about access to their article until it has been accepted. Find out more about Transformative Journals

Authors may need to take specific actions to achieve compliance with funder and institutional open access mandates. If your research is supported by a funder that requires immediate open access (e.g.

according to Plan S principles) then you should select the gold OA route, and we will direct you to the compliant route where possible. For authors selecting the subscription publication route, the journal's standard licensing terms will need to be accepted, including self-archiving policies. Those licensing terms will supersede any other terms that the author or any third party may assert apply to any version of the manuscript.

With best regards,

Giacomo Ariani
Editor
Nature Human Behaviour